# Nucleoplasmic signals promote directed transmembrane protein import simultaneously via multiple channels of nuclear pores

Krishna C. Mudumbi [1,2,3✉], Rafal Czapiewski [4], Andrew Ruba[1], Samuel L. Junod[1], Yichen Li[1], Wangxi Luo[1], Christina Ngo[1], Valentina Ospina[1], Eric C. Schirmer [4✉] & Weidong Yang [1✉]

Roughly 10% of eukaryotic transmembrane proteins are found on the nuclear membrane, yet how such proteins target and translocate to the nucleus remains in dispute. Most models propose transport through the nuclear pore complexes, but a central outstanding question is whether transit occurs through their central or peripheral channels. Using live-cell high-speed super-resolution single-molecule microscopy we could distinguish protein translocation through the central and peripheral channels, finding that most inner nuclear membrane proteins use only the peripheral channels, but some apparently extend intrinsically disordered domains containing nuclear localization signals into the central channel for directed nuclear transport. These nucleoplasmic signals are critical for central channel transport as their mutation blocks use of the central channels; however, the mutated proteins can still complete their translocation using only the peripheral channels, albeit at a reduced rate. Such proteins can still translocate using only the peripheral channels when central channel is blocked, but blocking the peripheral channels blocks translocation through both channels. This suggests that peripheral channel transport is the default mechanism that was adapted in evolution to include aspects of receptor-mediated central channel transport for directed trafficking of certain membrane proteins.

[1] Department of Biology, Temple University, Philadelphia, PA 19122, USA. [2] Department of Pharmacology, Yale University School of Medicine, New Haven, CT 06520, USA. [3] Yale Cancer Biology Institute, Yale University, West Haven, CT 06516, USA. [4] The Institute of Cell Biology, University of Edinburgh, Edinburgh EH9 3BF, UK. ✉email: krishna.mudumbi@yale.edu; e.schirmer@ed.ac.uk; weidong.yang@temple.edu

The nuclear envelope is composed of outer (ONM) and inner (INM) nuclear membranes that form a boundary between the nucleus and the cytoplasm. The ONM is contiguous with the endoplasmic reticulum (ER) in the cytoplasm and many ONM proteins have important interactions with the cytoskeleton[1–3], while the INM faces the nucleoplasm[4] and many INM proteins have important functions in genome regulation[5–7]. Thus, mechanisms for directed trafficking of transmembrane proteins into the nuclear compartment are critical for the cell. Selective bidirectional trafficking of soluble molecules between the nucleus and cytoplasm is mediated by nuclear pore complexes (NPCs) embedded where the ONM and INM fuse[8]. NPCs are megadalton macromolecular structures composed of ~30 different nucleoporins (Nups) arranged in eightfold rotational symmetry[9–11]. Trafficking of soluble proteins between the cytoplasm and the nucleus has been well studied and occurs through the central channel of the NPC[12–15], but the path taken by nuclear envelope transmembrane (NET) proteins into the nucleus remains in dispute[16–22].

Transit into the INM can, in theory, occur through either NPC-dependent or NPC-independent routes (Fig. 1a and Supplementary Fig. 1). Though NPC-independent transport has been seen in viral egress[23,24], no study has found evidence for its use in INM protein import. Both NPC-dependent and independent transport of NETs requires that NETs stay embedded in the membrane for the transit process to remain energetically favorable. For NPC-dependent transport, this suggests that transit might occur through ~10 nm wide peripheral channels of the NPC that were identified by early electron microscopy (EM) studies roughly 30 years ago[10,25,26]; however, the exact location, dimensions, and functionality of these channels has remained unclear[27–29]. Functional confirmation of this peripheral channel transport route is also of wide interest because many pathogens are known to disrupt central channel transport[30–36], and in such cases the peripheral channels could function as a critical backup mechanism for nuclear signaling.

Various models (Fig. 1a and Supplementary Fig. 1) have been proposed for how transmembrane proteins destined for the INM reach the nucleus[37–41]. This study assesses two NPC-dependent models: free lateral diffusion-retention and nuclear localization signal (NLS)-dependent facilitated transport. Both models require that the transmembrane domain of INM proteins stays embedded in the nuclear envelope during transit from the ONM to the INM, but there are some critical differences between the two mechanisms. In the lateral diffusion-retention model, transmembrane proteins freely diffuse in the membrane between the ONM and INM without obstacles, and directionality comes from retention by binding partners in the INM. In this model, INM proteins are restricted to the multiple peripheral NPC channels, which cryo-electron microscopy (cryo-EM) suggests are ~10 nm wide[11,25,42–44]. This would limit the nucleoplasmic domains of INM proteins to ~60 kDa if they had a globular structure since this would yield a hydrodynamic radius of ~10 nm. If their structure were more linear the proteins could still, in theory, snake through these channels with a smaller radius in the orientation of transport. This 60 kDa limit has been experimentally confirmed[17,45] and is characteristic for the wide range of NETs identified in the nuclear envelope[18,19]. However, most NETs are not likely to have a rigid highly folded structure as most have regions of intrinsic disorder (ID) within their nucleoplasmic domains. Several nucleoplasmic signals—signals such as nuclear localization signals and these ID regions found on the nucleoplasmic domain of NET proteins—have been shown to be important for transport. For instance, transport receptors that have been shown to facilitate central channel transport[46–49] are too big to fit into the peripheral NPC channels, yet transport receptors importin alpha and beta were shown to be important for NET transport in yeast[21,22]. Thus, the NLS-dependent mechanism posits that along with NLSs, long ID regions in the nucleoplasmic domains of INM proteins are required. These ID domains could, in theory, stretch through the core NPC structure to allow the NLS containing nucleoplasmic domain to reach into the central channel (~50 nm wide at the narrowest region), which enables the NLSs to bind transport receptors and phenylalanine-glycine (FG) Nups, and recapitulate transport similar to that of soluble proteins[50]. However, previous studies used truncated NET proteins, and the relevance of endogenous ID regions in full-length genome-encoded NETs remains to be tested.

Given the aforementioned dimensions of the NPC's central and peripheral channels, and the possible distance between them (~20–50 nm), three-dimensional (3D) super-resolution light microscopy could be appropriate to distinguish the transport of proteins through these channels in cells. Recently developed 3D super-resolution light microscopy techniques such as 3D-STORM, 3D-PALM, and 3D-STED have made breakthroughs in better understanding cellular mechanisms at the nanoscale. However, due to the required temporal and spatial resolutions for imaging nucleocytoplasmic transport in live cells (<10 nm and <1 ms)[51,52], these techniques might be inadequate to solve this problem. Therefore, to conquer these limitations, we have employed multiple advanced single-molecule techniques, including single-molecule fluorescence recovery after photobleaching (smFRAP)[53], single-molecule Förster resonance energy transfer (smFRET), and a high-speed virtual 3D single-molecule method termed single-point edge-excitation sub-diffraction (SPEED) microscopy[51,52]. Notably, SPEED microscopy combines high-speed 2D single-molecule localization microscopy and a post-localization 2D to 3D transformation algorithm, in which the former records single-molecule locations of transmembrane proteins through the NPCs with a spatiotemporal resolution of <10 nm and 0.4 ms, and the latter generates 3D spatial locations of transport routes for membrane proteins in the NPCs based on reconstructing recorded 2D single-molecule localizations.

Using these three approaches we have been able to distinguish protein domains translocating in the central and peripheral NPC channels and in the nuclear envelope lumen. Critically, this demonstrates that all NETs remain embedded in the membrane during transport and gives functional insights into the mechanism of transmembrane protein transport, revealing that only ~9% of the hundreds of INM NETs—based on the combined appearance of NLS and ID domains of appropriate length—should have the potential to use transport receptors. Such INM proteins have distinct domains in the central and peripheral channels during transport, allowing the domain in the central channel access to transport receptors. However, when central channel transport is blocked these INM NETs can revert to solely using the peripheral channels for transport.

## Results

**INM proteins use the NPC peripheral channels**. To test the two NPC-dependent transport route models (Fig. 1a and Supplementary Fig. 1), we selected several endogenous NETs with a range of characteristics to determine if different NETs use distinct transport routes, and if so, what characteristics direct the use of a particular route. Distinct INM protein candidates were selected based on the following criteria: (1) different sizes of extraluminal soluble domains (N and/or C terminus), (2) absence or presence of one or more predicted NLSs, and (3) absence or presence of endogenous extraluminal ID regions (Supplementary Table 1). By studying INM proteins with various combinations of these three criteria, we probed the requirements for predicted lateral

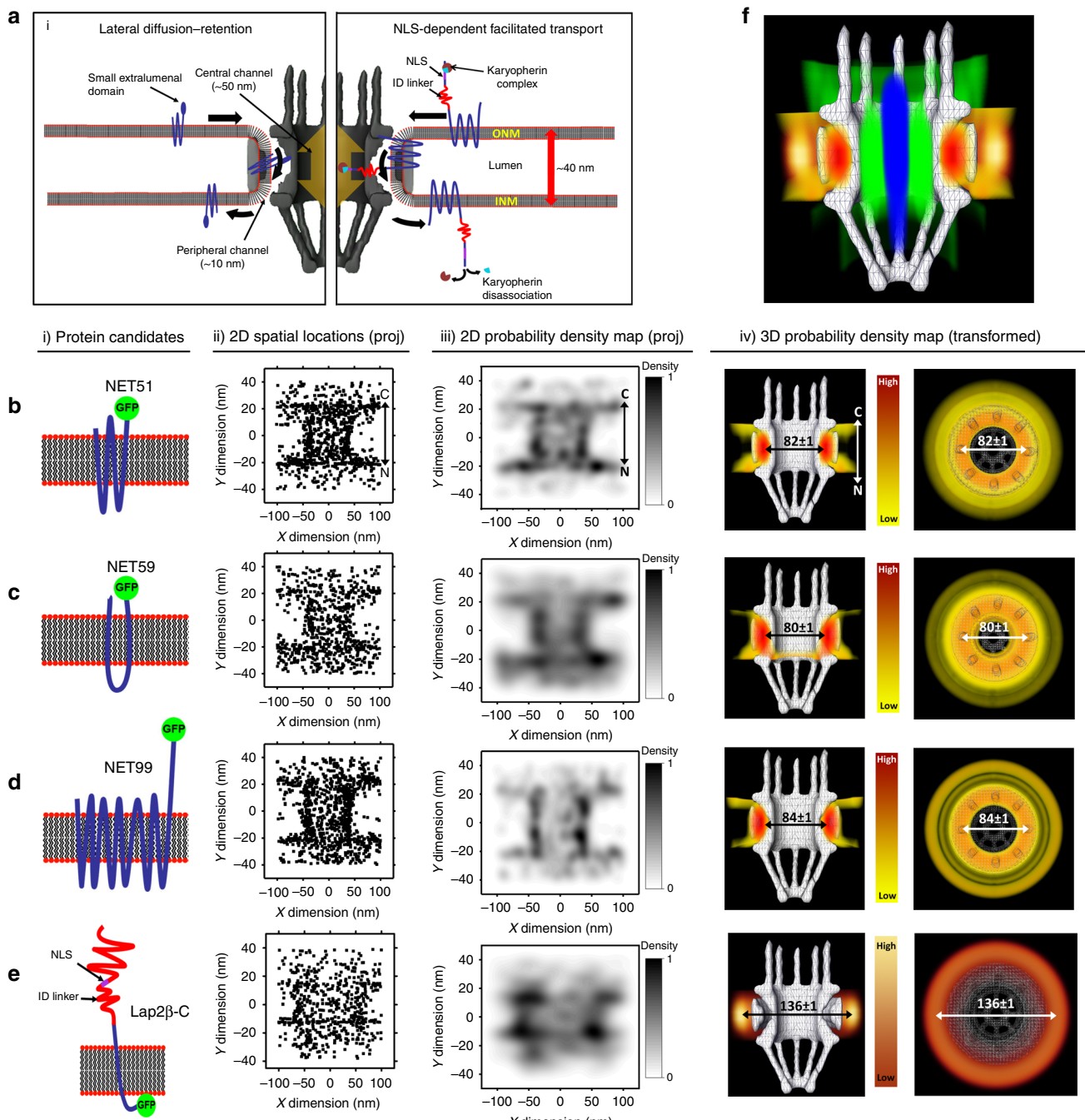

**Fig. 1 INM proteins use the NPC's peripheral channels to enter the nucleus. a** The two suggested NPC-dependent transport models for INM proteins (blue). (i) The lateral diffusion-retention model suggests that INM proteins with small luminal domains (blue) may freely diffuse through ~10 nm wide peripheral channels located in the physical structure of the NPC (dark gray). (ii) The NLS-dependent facilitated transport suggests that an extraluminal terminus of INM proteins containing an NLS and ID regions will be recognized and carried by karyopherin complexes (brown and light blue) through the NPC's central channel. **b-e** 3D spatial locations of transport routes for NET51, NET59, NET99, and Lap2β-C respectively, as determined by SPEED microscopy. (i) Schematic cartoon representation of each INM protein (orientation N-terminus to the left and C-terminus to the right). The depicted membrane represents a single bilayer of the NE. (ii) 2D spatial locations (projected) of each INM protein candidate. In the X dimension of the Cartesian graph, 0 represents the axial center of the NPC, and in the Y dimension, +20 represents the ONM, and −20 represents the INM. (iii) A normalized 2D probability density map showing the regions where the projected single-molecule localizations had the highest and lowest density. (iv) 3D probability density map generated by using the 2D to 3D transformation algorithm[52], with regions of highest to lowest density (red to yellow in color bar for NET51, NET59, and NET 99; light yellow to dark red for Lap2β-C) overlaid on the physical structure of the NPC (white mesh). **f** Merged view of the 3D spatial density maps of INM proteins using the peripheral channel (NET51, red to yellow) and nuclear envelope lumen (Lap2β-C, light yellow to dark red), small soluble molecules using the axial center (10-kDa dextran, blue)[51], and transport receptors using the central edge of the NPC (Importin-β1, green)[51].

diffusion-retention and NLS-dependent facilitated transport mechanisms.

First, three INM protein candidates lacking predicted NLSs and ID regions were analyzed. Each candidate had varying sizes for their extraluminal (e.g., cytoplasmic/nucleoplasmic) domains (NET51, 1.2 kDa; NET59, 2.3 kDa; NET99, 27 kDa) that were fused with EGFP (Fig. 1b[i]–d[i]) and then experimentally imaged by SPEED microscopy as they crossed the NPC (Supplementary Figs. 1, 2, and Supplementary Note 1). We typically collected 1000 2D single-molecule localizations (Supplementary Figs. 3 and 4) with a single-molecule localization precision <10 nm (Supplementary Fig. 5) from 5 to 10 cells for each protein candidate. We then applied post-localization 2D to 3D transformation algorithms to generate a spatial probability density map (Supplementary Figs. 6 and 7). The minimum number of localizations, the optimal bin size, and the single-molecule localization precision have been examined by computational simulations (Methods) to ensure a highly reliable 3D determination of spatial location for the transport route taken by each protein, with a route precision of <2 nm (Supplementary Table 2. and Supplementary Figs. 8–10). We found that NET51, NET59, and NET99 all have the highest spatial probability densities at a radial distance of ~41 nm from the central axis of the NPC (Fig. 1b[iv]–d[iv]). As transport routes for NLS-containing soluble proteins yield a radius of 23 ± 3 nm (50 in replicates) from the NPC's central axis[51,52] and the central channel has a measured radial width of ~25 nm by cryo-electron tomography[54], this indicates the NET transport routes stay well outside of the central channel of the NPC. Further supporting these structural dimensions, multiple cryo-EM studies have independently determined the position of the peripheral channels to be ~41 nm radial distance from the center of the NPC[42,55]—identical to our measurements. Moreover, others reported these peripheral channels at ~10 nm in diameter, similar to our measurement of ~12 nm full width at half maximum (FWHM, Supplementary Fig. 11). Thus, our measurements of protein translocation and NPC core proteins using SPEED microscopy are in strong agreement with measurements of the peripheral channels performed by cryo-EM, thus verifying that these proteins transport through the peripheral channel. We also verified that nuclear envelope curvature does not affect the determination of the 3D transport route. The pore membrane surrounding the NPC is curved and so the distance of the peripheral channel at the top of the NPC is slightly further away than the distance midway through the channel. Measurements using different bin sizes for the NPC resulted in nearly identical transport routes (Supplementary Table 3). Importantly, these results provide the first in vivo evidence that peripheral channels exist in the physical structure of the NPC, and their functionality in live cells.

These data are consistent with the lateral diffusion-retention mechanism for several additional reasons. The membrane topology of these NETs has been confirmed experimentally so that these nucleoplasmic regions reflect the only part physically capable of reaching the central channel. The proteins with the shorter nucleoplasmic domains—NET51 and NET 59—would not be long enough to stretch across the NPC core structure and into the central channel even if fully unfolded and stretched because there are too few amino acid residues beyond the membrane spanning regions and the GFP could not be stretched and still retain its fluorescence. Thus, these NETs serve as controls in effect for other NETs that, if stretched, could theoretically reach into the central channel, yet do not, such as NET99 which was also constrained to the peripheral channels. Further supporting the physical structural parameters of the NPC and pore membrane, we also tested the translocation pathway of

the C-terminus of lamina-associated polypeptide 2-beta (Lap2β-C), which has been confirmed in topology studies to protrude into the lumen of the nuclear envelope, and we show that it moves through the lumen with a route radius of 68 ± 1 nm (10 in replicates) (Fig. 1e). Again, the accuracy of these measurements is underscored by the corollary measurements made for the luminal domain by cryo-EM studies[42,55,56]. Our measurements from this independent approach in unfixed and unprocessed cells in turn argues that NPC structure and membrane positioning were not altered by processing in the earlier cryoEM studies[42,55,57]. This combination of data from both nucleoplasmically and luminally tagged INM proteins further confirms aspects of the lateral-diffusion retention model (Fig. 1f, Supplementary Movie 1).

**Some nucleoplasmic domains transit the central channel.** One variant of the NLS-dependent facilitated transport model argues that both NLSs and ID regions are required for transport[22]. Thus, next we determined the route used by two INM proteins containing strong NLSs and ID regions in their extraluminal soluble domains that were long enough in theory to reach through the ~20-nm physical scaffolding of the NPC[55]. The thickness of the physical scaffolding, i.e., the distance between the central and peripheral channels, was determined by the single-molecule experiments and cryo-EM explained above, and the number of amino acids estimated to be required to span this length was calculated using an average amino acid backbone length of 0.45 nm as has been measured previously in stretched peptides[58,59] (explained in further detail in the Supplemental Methods). Two well characterized INM proteins met these requirements: lamin-B receptor (LBR; Fig. 2a[i]–b[i]) and Lap2β (Fig. 2c[i]), with respectively the longest extraluminal domain being 208 or 410 aa. The C-terminus of LBR is 39 aa long and extraluminal while the C-terminal region of Lap2β is 24 aa and in the nuclear envelope lumen. Both proteins were tagged with EGFP on the NLS-containing domain (LBR-N, Lap2β-N) and observed via SPEED microscopy. The 3D spatial reconstruction shows the LBR-N amino-terminal EGFP moiety mainly moves through a route located at a radial distance of 25 ± 1 nm from the center of the NPC (Fig. 2a[ii–iv]). Similarly, the EGFP tagged Lap2β-N had a radial distance of 24 ± 1 nm from the NPC's central axis (Fig. 2c[ii–iv]). Given that the NPC has a ~25 nm radius at its narrowest point, the above measurements suggest that both LBR-N and Lap2β-N go through the edge of the NPC's central channel, following a similar transport route as previously determined for soluble proteins containing NLSs[51,52] and physically capable of binding their NLSs with central channel transport receptors.

LBR is a multispanning transmembrane protein with 8 transmembrane segments where both the N- and C-termini are extraluminal. However, the C-terminus is much shorter and lacks an ID region, and therefore is expected to stay in the peripheral channel. Indeed, tagging LBR on its C-terminal domain (LBR-C) with EGFP revealed that its C-terminus moves through the peripheral channel with a route radius of 41 ± 1 nm (Fig. 2B). This indicates that INM protein domains with a sufficiently long ID region and NLS can use the central channel while the transmembrane domains are still embedded in the membrane during transit (Fig. 2d and Supplementary Fig. 12).

To directly test that the LBR N-terminus transports in the central channel simultaneously with the same LBR C-terminus transporting through the peripheral channel, we used dual-channel co-tracking SPEED microscopy of LBR tagged with GFP on the N-terminus and RFP on the C-terminus (GFP-LBR-RFP). This confirmed that each LBR protein can simultaneously access both the central (route radius of 25 ± 1 nm) and peripheral channels (route radius of 44 ± 2 nm) during transport through the

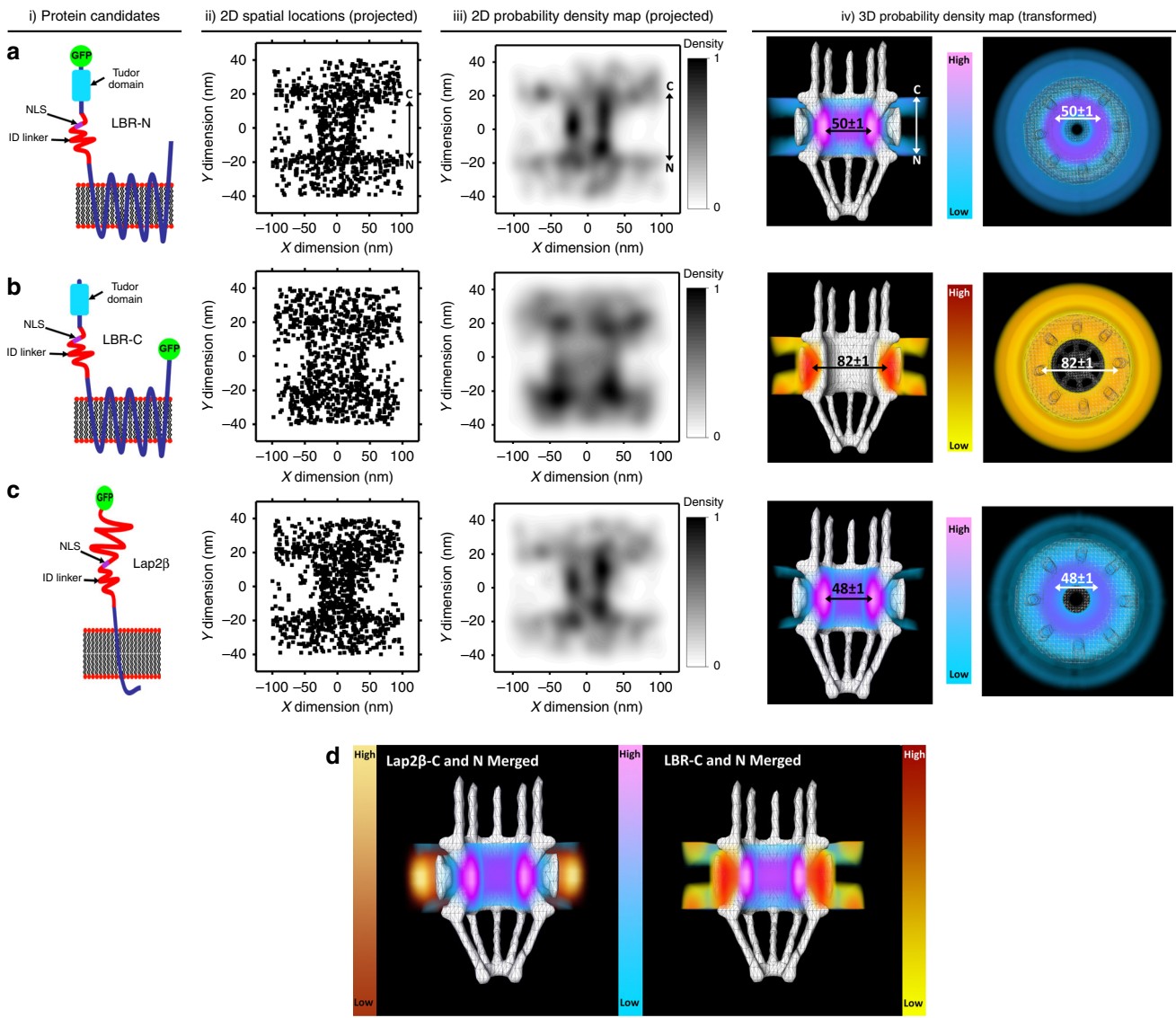

**Fig. 2 Translocation routes of LBR and Lap2β from the ONM to INM in live cells. a–c** Transport routes of N-terminally tagged LBR (LBR-N), C-terminally tagged LBR (LBR-C), and N-terminally tagged Lap2β (Lap2β-N) respectively. (i) Schematic cartoon representation of each INM protein (orientation N-terminus to the left and C-terminus to the right). The depicted membrane represents a single bilayer of the NE. (ii) 2D spatial locations (projected) of each INM protein candidate. In the X dimension of the Cartesian graph, 0 represents the axial center of the NPC, and in the Y dimension, +20 represents the ONM, and −20 represents the INM. (iii) A normalized 2D probability density map showing the regions where the projected single-molecule localizations had the highest and lowest density. (iv) 3D probability density map generated by using the 2D to 3D transformation algorithm[52], with regions of highest to lowest density (each color bar shows the gradient color change from high to low density). **d** Side by side comparison of the 3D probability density map of Lap2β N and C terminus with LBR N and C terminus. Color bars indicate the regions of highest and lowest density.

NPC (Fig. 3, Supplementary Movie 2 and Supplementary Movie 3). Furthermore, since GFP and RFP are FRET pairs, we performed experiments in which the GFP was excited and the RFP channel was observed for FRET. These experiments did not produce a FRET signal, indicating that the GFP and RFP did not both travel through the peripheral channel, where they would be closely confined and give a FRET signal. This further solidifies the argument that termini containing both an NLS and ID region follow a different transport pathway than termini which lack these elements.

To interrogate the relative contribution of the ID domains and NLSs of LBR in the transport process, various mutations were engineered for the EGFP tagged N-terminus and imaged using SPEED microscopy. First, a 110 amino acid region, spanning from amino acid 63 to 172, was removed (LBR Δ63-172) so that LBR would no longer have either an ID domain or NLS. The

removal of both the ID region and NLS resulted in the N-terminus of LBR using the peripheral channels in the NPC to enter the INM, with a route radius of 39 ± 1 nm (10 in replicates) (Fig. 4a). Note that enough N-terminal LBR residues remained in this construct to in theory reach into the central channel of the NPC if stretched. The second construct was a point mutation in just one of the four LBR NLS sequences, where an arginine at position 74 of the strongest predicted NLS—also supported by previous studies[60]—was mutated to a threonine (LBR R74T), which should, in theory, create a non-functional NLS while keeping the ID domain intact and the total length of the N-terminus unchanged. The N-terminus of LBR R74T also used the peripheral channel route to enter the nucleus with a route radius of 40 ± 1 nm (10 in replicates) (Fig. 4b), indicating that the mutation of just this one NLS was sufficient to change the transport pathway of LBR. Similarly, the NLS in the N-terminus

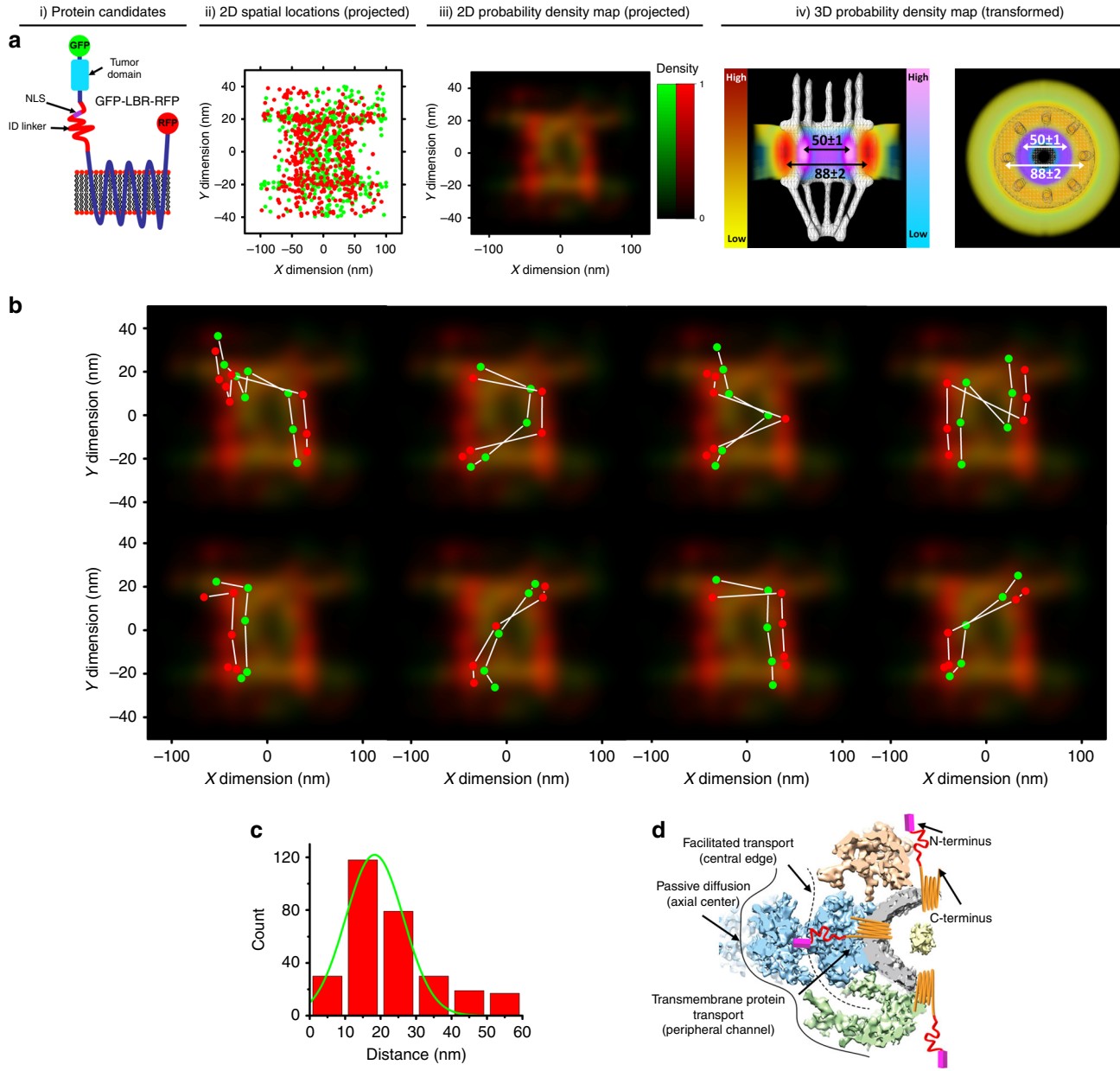

**Fig. 3 Simultaneous dual-channel single-molecule tracking of both termini of LBR. a** Transport route of LBR tagged with EGFP on the N terminus and RFP on the C terminus (GFP-LBR-RFP). (i) Schematic cartoon representation of each INM protein (orientation N-terminus to the left and C-terminus to the right). The depicted membrane represents a single bilayer of the NE. (ii) 2D spatial locations (projected) of each INM protein candidate. In the X dimension of the Cartesian graph, 0 represents the axial center of the NPC, and in the Y dimension, +20 represents the ONM, and −20 represents the INM. (iii) A normalized 2D probability density map showing the regions where the projected single-molecule localizations had the highest and lowest density. (iv) 3D probability density map generated by using the 2D to 3D transformation algorithm[52], with regions of highest to lowest density (each color bar shows the gradient color change between regions of high and low density). **b** Typical co-tracking trajectories of GFP-LBR-RFP in the NPC show that the EGFP and RFP tagged termini move in the same direction with close proximity to each other. Green dots represent EGFP, red dots represent RFP. These trajectories have been overlaid on top of the 2D probability density map of GFP-LBR-RFP. **c** Histogram of ~300 different dual tracked trajectories shows that the average distance between EGFP tagged N terminus and RFP tagged C terminus is ~18 nm during transit through the NPC. No FRET was observed during these experiments further confirming that the N and C termini do not transit in the same channel. **d** Cryo-EM image of the NPC with a cartoon representation of how LBR transits from the ONM to INM (figure modified from ref. [55] with permission).

of Lap2β was mutated (two arginines were mutated to threonines, Lap2β ΔNLS), and this changed its transport route from having a radial distance of 24 ± 1 nm from the NPC's central axis to a radial distance of 41 ± 1 nm (10 in replicates) from the NPC's central axis (Fig. 4c). Finally, we reduced the length of just the ID region of LBR by deleting 42 amino acids from the N-terminus (LBR ΔLinker; several amino acid regions within ID sequences were removed to keep all NLSs intact: 50–62, 80–94, and

112–125). It is interesting to note that the route radius for LBR ΔLinker changed from 25 ± 1 nm (LBR-N) to 39 ± 1 nm (10 in replicates), which demonstrates that even with four intact predicted NLSs and roughly 80% of the length of the nucleoplasmic region still present, LBR requires these ID regions to transit through the central channel of the NPC (Fig. 4d).

We next used these tested properties to predict the transport routes of other confirmed INM NETs. First, we collected

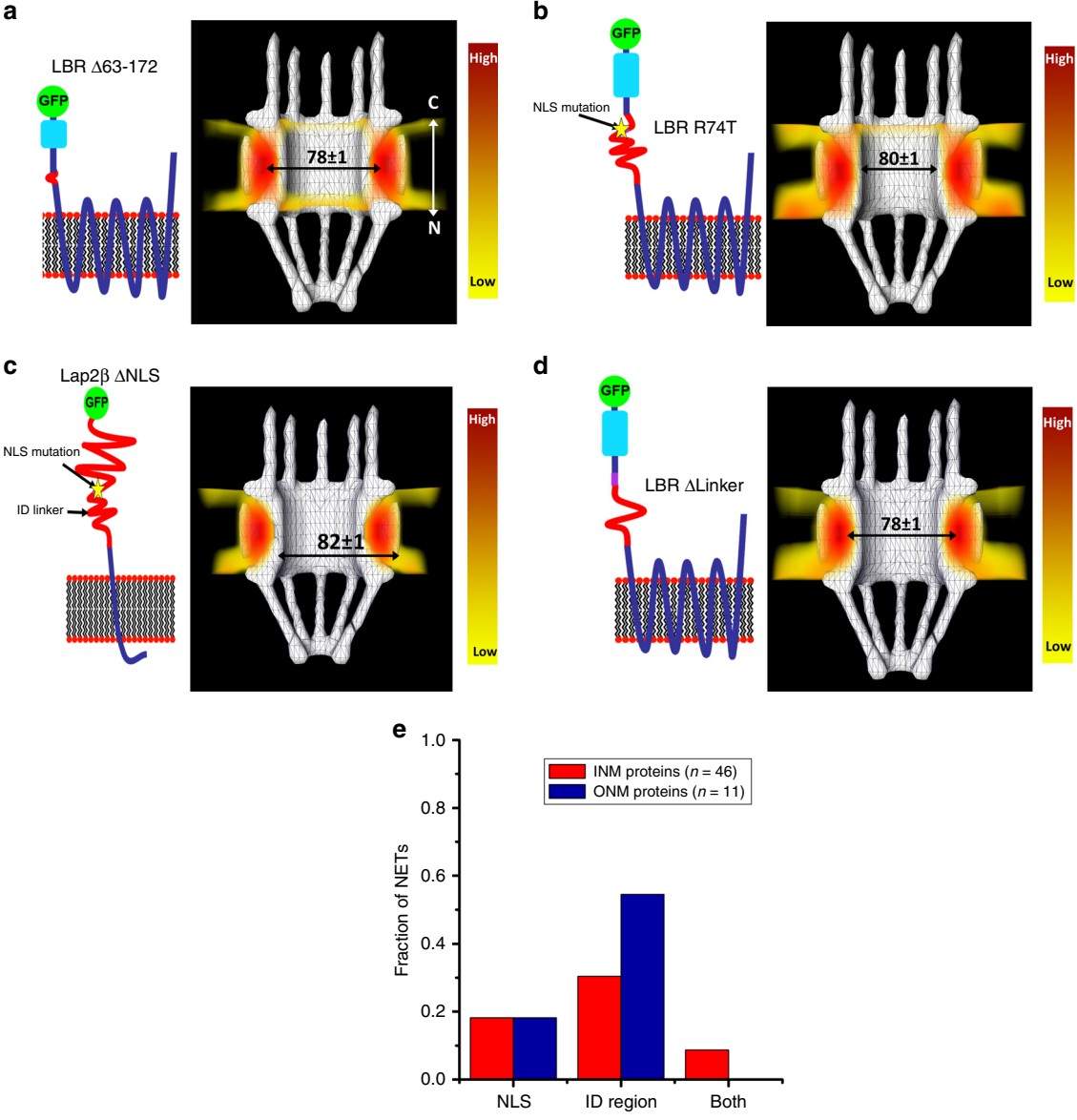

**Fig. 4 In vivo transport routes for mutated LBR constructs. a–d** Transport routes of N-terminally tagged LBR Δ63-172, LBR R74T, Lap2β ΔNLS, and LBR ΔLinker respectively. On the left, schematic cartoon representation of each INM protein (orientation N-terminus to the left and C-terminus to the right). The depicted membrane represents a single bilayer of the NE. On the right, 3D probability density map generated by using the 2D to 3D transformation algorithm[52], with regions of highest to lowest density (red to yellow in color bar). **e** Bioinformatic analysis of the fraction of confirmed INM proteins (red, $n = 46$) and ONM proteins (blue, $n = 11$) that have a strong NLS sequence, an ID region longer than 65 amino acids, and both of those characteristics arranged in a way that is feasible for transport through the facilitated transport route using both central and peripheral channels of the NPC.

information on NET targeting from our own and other published studies, assembling a dataset of confirmed INM ($n = 46$) and ONM ($n = 11$) localizing proteins (Supplementary Table 4). We then used bioinformatics to see what fraction of these proteins have these demonstrated characteristics that would allow them to use the central channel transport route. The aforementioned characteristics include: an NLS with a prediction strength of at least 0.86 (a strong predictor as determined by the SeqNLS software), and a long enough predicted ID region—>65 amino acids based on an average amino acid backbone length of 0.45 nm when unfolded and stretched[58,59]—that could stretch across the roughly 20 nm of the NPC core particle (further details given in Supplementary Methods). Our analysis predicts that only ~9% of INM proteins and no ONM proteins have both a strong NLS sequence and long enough ID region arranged in a way that allows for transport through the central channel of the NPC (Fig. 4e).

**NLS and ID regions enhance transport rates**. Since both the NLS and ΔLinker mutations altered the route of the N-terminus of LBR from the central channel to the peripheral channels, we examined how these changes affected the transport rate of LBR from the ONM to INM by using smFRAP[53]. This is a recently developed technique that provides quantitative information about the translocation rate (TR) of INM proteins in live cells. Using smFRAP, we observed that the normal TR of unaltered LBR from the ONM to INM is ~5.6 molecules per minute per NPC (Table 1, Supplementary Table 5, and Supplementary Fig. 13), which is in good agreement with previous studies[20,53]. However, the constructs LBR R74T and LBR ΔLinker changed the transport rate to ~2.9 and ~3.2 molecules per minute per NPC, corresponding respectively to a ~48% and a ~43% decrease in TR compared to the unchanged LBR. Thus, both the NLS and ID domain play a critical role in determining the TR of membrane proteins into the nucleus.

**Table 1 Determined translocation rates for LBR and its mutants.**

| Protein | Single-molecule based concentration ratio (INM:ONM) | Diffusion coefficient on ONM ($\mu m^2 s^{-1}$) | Diffusion coefficient on INM ($\mu m^2 s^{-1}$) | Diffusion based corrected concentration ratio (INM:ONM) | Immobilized fraction | Final concentration (INM:ONM) | Transport rate ([mol] min$^{-1}$NPC$^{-1}$) |
|---|---|---|---|---|---|---|---|
| LBR-N | 0.53:1 | 2.6 ± 0.8 | 1.9 ± 0.6 | 0.58:1 | 60 ± 6% (overall) 80 ± 6% (INM) | 3(±0.03):1 | 5.6 ± 0.3 |
| LBR R74T | 0.53:1 | 2.4 ± 0.6 | 2.0 ± 0.4 | 0.58:1 | 59 ± 5% (overall) 80 ± 5% (INM) | 2.9(±0.03):1 | 2.9 ± 0.3* |
| LBR ΔLinker | 0.41:1 | 1.1 ± 0.7 | 2.5 ± 0.5 | 0.48:1 | 30 ± 4% (overall) 57 ± 4% (INM) | 1.1(±0.03):1 | 3.2 ± 0.3* |

Using the previously described smFRAP method[53], the ONM:INM distribution ratios for LBR, LBR R74T and LBR ΔLinker were determined. These values were then adjusted by using the diffusion coefficients on the ONM and INM (determined through single-molecule tracking microscopy) and the immobilized fraction (determined through ensemble FRAP). The corrected values were then used to determine the transport rates for these proteins as molecules per minute per NPC ([mol]min$^{-1}$NPC$^{-1}$). A one-way ANOVA with a Tukey's HSD post hoc test was used to show a significance between LBR R74T and LBR ΔLinker as compared to LBR-N with $p = 1 \times 10^{-12}$, $F = 179$, df = 2.

**The peripheral channels are essential for INM transport**. To test the relative functional importance of the NPC's central and peripheral channels in nuclear translocation of transmembrane proteins, we performed experiments imaging the nuclear transport of LBR-N after disrupting either central or peripheral channel transport. To block the central channel we microinjected wheat germ agglutinin (WGA), a lectin that has previously been shown to bind O-glycosylated nucleoporins and dramatically reduce transport through the central channel[17,61,62]. To block the peripheral channels we microinjected an antibody to the region of the transmembrane nucleoporin gp210 (anti-gp210) that reaches from the membrane to the central mass. This has been previously reported to block transport of reporter transmembrane constructs, presumably by blocking transport through the peripheral channels[17]. We found that microinjection of WGA prevented the wild-type LBR-N from using its normal route through the central channel of the NPC to enter the nucleus. Instead, the protein was re-directed to just use the peripheral channels, as indicated by the highest density of LBR molecules diffusing through the NPC with a radial distance of 43 ± 1 nm (10 in replicates) from the NPC's central axis (Fig. 5a, c, f) as compared to the 25 ± 1 nm (10 in replicates) radial distance of wild-type LBR-N in untreated cells (Fig. 5e). It is unclear if these transiting LBR molecules, having their NLSs intact, are still associated with importin-α/β complexes; however, it seems unlikely as the importins are in theory too large to transit through the peripheral channels.

The injection of anti-gp210 antibodies greatly reduced the probability of LBR-N to complete its translocation through either the NPC's peripheral or central channels, despite that this treatment has no inhibitory effect on the translocation of soluble molecules through the NPC's central channel. The block in LBR translocation with the gp210 antibodies is evidenced by the reduction in LBR at the inner aspect of the NPC, shown in both 2D histogram and transformed 3D probability density maps (Fig. 5b, d, and g). Additionally, compared to LBR's distribution on the ONM and INM in the absence of WGA and anti-gp210, the addition of these reagents increases the concentration of LBR on the ONM compared to the INM (Fig. 5e–g), and anti-gp210 causes a much more pronounced bias between ONM and INM accumulation than WGA.

## Discussion

In this study we answered some of the most debated fundamental questions remaining in nucleocytoplasmic transport of transmembrane proteins. First, by using in vivo imaging, we showed the existence of functional peripheral channels in the NPC, and how they are used by INM proteins to diffuse from the ONM into the INM. This supports earlier EM indications of peripheral channels[11,25,39,41,63]. Next, we provided evidence that the nucleoplasmic domains of INM proteins containing both NLS (s) and ID regions use the central channel and follow a similar route as soluble NLS-containing proteins; meanwhile the transmembrane domains of these INM proteins remain anchored in the nuclear envelope and travel through the peripheral channel of the NPC. Notably, disruption of these 'transport sequences' not only changed the transport mechanism from the central to the peripheral channel for the nucleoplasmic domains of INM proteins, but also lowered TRs of these INM proteins from the ONM to the INM.

Several recent studies also used in vivo imaging techniques, which took advantage of cleavable sequences to determine INM protein transport[16,20]. Using these methods, both groups concluded that LBR and Lap2β transit into the INM using some form of lateral-diffusion retention. Ungricht and colleagues also conclude that this transit is energy dependent, which is suggestive of transport receptor mediated trafficking. In contrast, based on the INM localization of LBR after knocking down numerous transport receptors, Boni et al suggest that the lateral-diffusion retention model is sufficient to explain INM protein import since the presence of transport receptors do not seem to affect LBR localization. However, as shown by our approach, the loss of NLSs—equivalent to the loss of access to transport receptors—reduces the transport rate of LBR and changes its transport mechanism from the central channel of the NPC to the peripheral channel, but allows it to still localize to the INM. By directly observing INM protein transport and distinguishing between the lateral-diffusion retention and transport receptor mediated models, we could validate both approaches while providing more nuanced context.

The finding that different regions of the same protein transit simultaneously through both central and peripheral channels enhances previous studies—arguing for central channel transport[22,64]. In addition, our bioinformatic analysis indicates that, even though the majority of membrane proteins likely only use the NPC's peripheral channels for translocation, roughly a tenth of INM proteins could adopt the combined peripheral and central channel transport mechanism. Importantly, this subset of INM proteins has likely adapted this more complex transport mechanism for more efficient and directed trafficking into the

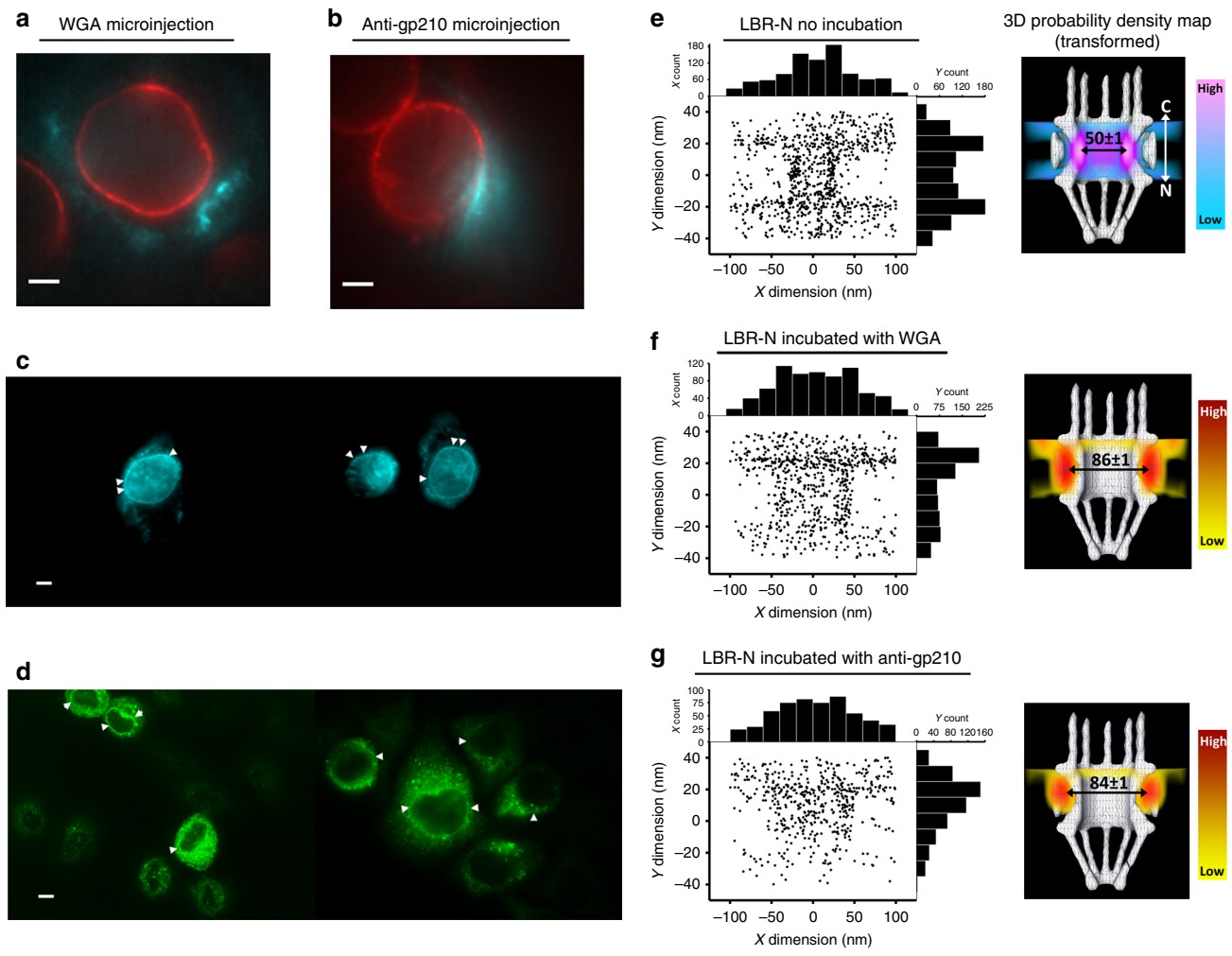

**Fig. 5 Transport routes for unaltered LBR with the presence of WGA or anti-gp210. a**, **b** HeLa cells expressing PoM121-mCherry (red) and LBR-N (GFP, not shown) immediately after microinjection with WGA conjugated to Alexa Fluor-647 (cyan) or a mixture of anti-gp210 and Alexa Fluor-647 (cyan, scale bar: 5 µm; at least eight biological replicates). **c** Fluorescent image of HeLa cells incubated with WGA (cyan) for 30 min (scale bar: 5 µm; at least three biological replicates). **d** Immunofluorescent image of HeLa cells after permeabilization and incubation with anti-gp210 (Alexa-488 conjugated secondary antibody, green) for 30 min (scale bar: 5 µm, at least three biological replicates). **e–g** On the left, projected 2D spatial localizations of unaltered LBR in live cells without and with microinjection of WGA or anti-gp210 (top to bottom). Histograms showing the number of LBR's locations in the x (top) and y (right) dimensions. On the right, 3D probability density map generated as previously mentioned (each color bar shows the gradient color change between regions of high and low density).

nucleus as evidenced by the faster transport rates of LBR when the protein was capable of reaching into the central channel. Notably, LBR and Lap2β are amongst the group of NETs that give the strongest nuclear rim staining with very small pools of the proteins in the ER, consistent with operation of a more directed transport mechanism. In addition, our analysis suggests a similar transport route for the NETs emerin and Lap1, both of which have vitally important cellular functions[65–71]. This is further supported by mutations in the intrinsically disordered domain of emerin, which have been shown to cause Emery-Dreifuss muscular dystrophy 1[72–74]. Furthermore, our use of live cell imaging clearly shows that different INM proteins use distinct transport routes through the NPC and that each route may have a distinct transport rate into the nucleus, suggesting that this range of transport routes might contribute to key aspects of regulating cellular homeostasis.

Critically, this study resolves a debate that has been going on for many years about the nature and even existence of the peripheral channels and arguments from different groups about their

precise dimensions in the NPC and how they integrate with the pore membrane. The dimensions we have measured for a protein domain in the lumen versus the peripheral channel clearly match those originally measured by cryoEM nearly 30 years ago. Furthermore, the spacing of the central channel against these other positions matches the cryoEM measurements (Fig. 6), so that both approaches yield similar dimensional parameters. This argues both for validating the SPEED microscopy approach and that cryo-EM techniques properly preserve the NPC and membrane structure.

The ability to change the transport route, but retain transport to the nucleus, by altering NLS and ID regions could explain why some INM protein disease alleles mistarget away from the INM. For example, emerin, a NET mutated in Emery-Dreifuss muscular dystrophy, binds to lamins for retention in the INM once it is translocated[75]. Many emerin disease alleles mistarget and accumulate in the ER[76–79]. Some disease alleles of emerin have been shown to have weaker binding to lamins, but for others this cannot explain the ER accumulation; however, if these mutations

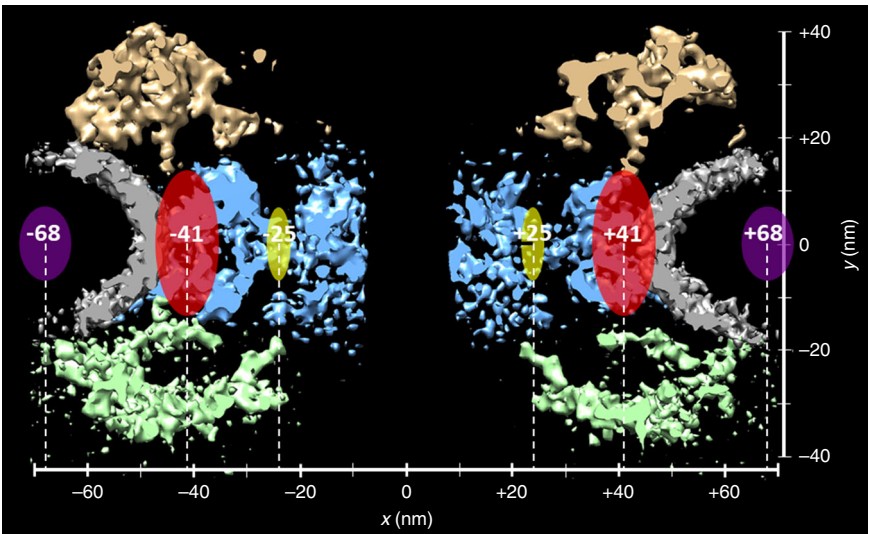

**Fig. 6 Transport route overlap on the NPC physical structure.** Single-molecule determined transport routes (purple, red, and yellow ovals) overlaid on cryo-EM determined NPC structure (tan, blue, green, gray; cryo-EM accession numbers: EMD-3005, EMD-3007, EMD-3009, EMD-3011). The C-terminus of Lap2β has been experimentally determined in previous studies to extend into the nuclear envelope lumen, and was measured in our work to have a translocation distance of 68 nm from the central axis of the NPC (purple). Correspondingly, the extraluminal domains of NETs and the mass proximal to the transmembrane regions transit at roughly 41 nm from the central axis of the NPC (red) where the peripheral channels have been observed by cryo-EM tomography. However, extraluminal NET regions with long ID regions and NLSs can transit at 25 nm from the central axis of the NPC (yellow), corresponding with central channel transport.

interfere with the ID or NLS of emerin, then this could be explained by our data. Lastly, this also suggests a protective built-in redundancy for nucleocytoplasmic transport as membrane proteins can still use the peripheral channels when the central channel is blocked, for example by viral infection. This would further underscore critical importance of proper membrane protein trafficking in evolution.

## Methods

**Tissue culture and transfection.** Both wild-type HeLa (ATCC) and stably transfected HeLa cell lines that express mCherry tagged PoM121 proteins were used in our experiments. These cells were grown in DMEM (Thermo Fisher) containing high glucose (Thermo Fischer), GlutaMAX Supplement (Life Technologies), 10% fetal bovine serum (Fischer Scientific), and 1% penicillin-streptomycin (Thermo Fischer). By following the manufacturer's protocol, we conducted cell transfection via electroporation (Bio-Rad GenePulser Xcell). Cells were incubated with pre-warmed (37 °C) transport buffer for 45 min before either single-molecule (SPEED and smFRAP) or ensemble FRAP experiments. Transport buffer contains 20 mM HEPES, 110 mM KOAc, 5 mM NaOAc, 2 mM MgOAc, 1 mM EGTA, and HCl for adjusting pH to 7.3. Microinjection experiments were performed with an Eppendorf FemtoJet 4I and InjectMan 4 using an injection pressure of 250 hPa, 65 hPa compensation pressure, and an injection time of 0.1 s. Cells were injected with either WGA (ThermoFisher, W32466, at 1.33 µM) or anti-gp210 (Novus Biologicals, NB100-93336, at 1.33 µM) and incubated for 30 min at 37 °C in transport buffer before imaging. Rabbit primary antibody anti-GFP (A-11122) and mouse anti-γ-tubulin (MA1-850) were purchased from Thermo Fisher. For Western blotting IR680 and IR800 conjugated goat anti-rabbit antibodies (LI-COR Biosciences) were used.

**Plasmids.** Human LBR clone was a gift from Professor Glenn Morris, Keele University, UK. Wild-type human Lap2B was a gift from Professor Roland Foisner, Medical University of Vienna, Austria. IMAGE clones for human NETs were obtained from RZPD and Geneservice. IDs are in brackets following the gene name. N-tagged proteins were cloned into pEGFP-C3 mammalian expression plasmid. C-tagged proteins were cloned into pEGFP-N2. Wild-type LBR N-tagged was cloned by XhoI and BamHI. LBR C-tagged was cloned by NheI and BamHI sites. LBR Δ63-172, NLS mutant R74T, and ΔLinker (Δ50-62, 80–94, 112–125) were generated by Quick Change mutagenesis. Wild-type and mutant human Lap2β was cloned by SalI and BamHI sites (N-tagged) ΔNLS mutation was obtained by Quick Change mutagenesis to replace two arginine sites to threonine R319T and R320T. NET51 (IMAGE 347242) was PCR amplified from its IMAGE clone and restriction sites XhoI/BamHI added (N-tagged). NET59 (IMAGE 3959506) was PCR amplified from its IMAGE clone and digestion sites NheI and

BamHI were added (C-tagged). NET99 (STT3A) (IMAGE 3891543) was cloned by ScaI and Bsp120I sites added during PCR from its IMAGE clone (N-tagged).

**Primers.** NET99
  F: cagtcgaccatatgactaagtttggattttttgcg
  R: gagggcccttatgtccttgacaagcctcg
NET51
  F: cactcgagcatatgagccgtttcctgaatgtg
  R: gaggatcctcagtttctcttcttctgtctgg
LBR
  F: cactcgagcatatgccaagtaggaaatttgcc
  R: gaggatccttagtagatgtatgggaaatatacggtaggg
Lap2B
  F: gtcgactatgccggagttcctggaagac
  R: ggatccattcagttggattttctagggtc
Lap2B ΔNLS
  F: gtttctttggtgctgttgtgggtatgtctgagaattcagtgat
  R: atcactgaattctcagacatacccacaacagcaccaaagaaac
LBR ΔLinker
  F1: ggaacagagcttgaattgaggaaaggtggctcaact
  R1: agttgagccacctttcctcaattcaagctctgttcc
  F2: acgccgagggagtcgacgccgatctg
  R2: cagatcggcgtcgactccctcggcgt
  F3: atctgctgatgctatttcccctccttgcttccttaatg
  R3: cattaaggaagcaaggaggggaaatagcatcagcagat
LBR ΔNLS
  F: ttccagttcccctccacacgccgagg
  R: cctcggcgtgtggaaggggaactggaa
RFP_LBR_GFP
  RFP_F BamHI: ggatccaccggtcgcc
  RFP_R BglII: agatctctacaggaacaggtggtggcg

**Optical setup of SPEED microscopy.** We used an Olympus IX81 microscope for cell imaging. The microscope was equipped with a ×100 oil-immersion objective (UPLSAPO 100XO, Olympus, 1.4 numerical aperture) and with an on-chip multiplication gain CCD camera (Cascade 128+, Roper Scientific). A 50-mW solid-state 488-nm laser (Obis) and a 50-mW solid-state 561-nm laser (Obis) were used to excite EGFP tagged transmembrane proteins and excite the mCherry-Pom121 labeling NPCs in HeLa cell lines respectively. An on-off laser excitation was generated by using a Newport optical chopper. In our setup, a dichroic filter (Di01-405/488/561/645-25 × 36, Semrock), an emission filter (NF01-405/488/561/635-25×5.0, Semrock), and two neutral density filters (Newport) were used. For data acquisition and processing, the Slidebook software package (Intelligent Imaging Innovations) was used. Single-molecule localization data was analyzed with the GLIMPSE software package (courtesy of the Gelles Lab).

**Technical features of SPEED microscopy**. To capture single transiting molecules through sub-diffraction-limit NPCs with high spatial (<10 nm) and temporal (<1 ms) resolution, our lab has previously developed single-point edge-excitation sub-diffraction (SPEED) microscopy. To achieve these high resolutions, we have conducted four main technical modifications on conventional epi-fluorescence microscopy. First, a zero-mode laser excitation beam passed through the center of the objective to form a vertical illumination point spread function (iPSF) in the focal plane of the objective. If an inclined iPSF is needed in the focal plane, the laser excitation beam was shifted off the center of the objective. The diffraction-limited iPSF allowed us to use a high detection speed (up to 0.2 ms per frame for the CCD camera used: Cascade 128+, Roper Scientific) via reducing the number of camera pixels required for detection. Meanwhile, the small iPSF significantly avoided out-of-focus fluorescence, particularly by using the inclined iPSF. Second, a high optical density (100–500 kW/cm²) in the small iPSF generated a high number of photons from the fluorophores within the millisecond detection time. Third, a collection of a high-number of photons in a short time period greatly reduced the negative effects of molecular diffusion on the single-molecule localization precision of moving molecules, resulting in a high spatial resolution. Finally, pinpointed illumination in live cells greatly reduced photo-induced toxicity. Thus, SPEED microscopy meets the needs of high-speed single-molecule tracking of macro-molecule through native NPCs with high spatial resolution.

**Localization precision of isolated fluorescent spots**. The localization precision for single fluorescent molecules was defined as the precision of determining the central point of each detected fluorescent diffraction-limited spot. Typically, a 2D Gaussian function was used to fit for the central point of fluorescent spot and the standard deviation of multiple measurements of the central point represented the localization precision for immobilized molecules. While, the determination of localization precision for moving molecules must include the influence of particle motion during image acquisition. In detail, the localization precision for moving substrates ($\sigma$) was determined by an algorithm of:

$$\sigma = \sqrt{F\left[\frac{16\left(s^2 + \frac{a^2}{12}\right)}{9N} + \frac{8\pi b^2\left(s^2 + \frac{a^2}{12}\right)^2}{a^2 N^2}\right]} \quad (1)$$

where $F = 2$, $N$ is the number of collected photons, $a$ is the effective pixel size of the detector, $b$ is the standard deviation of the background in photons per pixel, and

$$s = \sqrt{s_0^2 + 1/3D\Delta t} \quad (2)$$

$s_0$ is the standard deviation of the PSF in the focal plane, $D$ is the diffusion coefficient of the substrate and $\Delta t$ is the image acquisition time. In our experiments, the molecules with >2000 signal photons and in-focus Gaussian widths (0.5–1.0 pixel, corresponding to single GFP molecules located in the focal plane) were selected. Based on the above equations and the parameters determined experimentally ($N > 2000$, $a = 240$ nm, $b \approx 2$, $s_0 = 150 \pm 50$ nm, $D$ is in the rage of 1–3 $\mu m^2 s^{-1}$ for the tested substrates), thus, the localization precision of these molecules was determined to be <10 nm.

**Determination of the diffusion coefficients for INM proteins**. In our measurements, we have used two complementary approaches to determine the diffusion coefficients for INM proteins. First, we used the typical mean squared displacement approach (MSD, MSD = 4Dt for 2D trajectories) if single-molecule trajectories of a protein molecule consist of more than 6 frames. Second, if there are two to six consecutive frames obtained, we utilized the frequency distribution probability function,

$$\rho(\delta, t, D) = \left(\frac{\delta}{2Dt}\right)e^{\left(-\frac{\delta^2}{4Dt}\right)} \quad (3)$$

where $\delta$, $t$, and $D$ are the displacement between consecutive frames, the interval time, and the diffusion coefficient, respectively. We have collected approximately 50 single-molecule long trajectories that were analyzed by the first approach and more than 500 events analyzed by the second approach. Finally, an averaged diffusion coefficient was calculated for each INM protein.

**Determination of the centroid of the NPC**. First, a two-second exposure time image is taken of the POM121-mCherry tagged NPC and fit with 2D Gaussian functions to find the centroid of the NPC. Next, single-molecule images are taken of fluorescently tagged INM proteins. Both data are overlaid on top of each other, and the centroid of the NPC determined in the previous step is used to center (0,0) the data on a Cartesian coordinate system. From here, the single-molecule data of the INM proteins is precisely selected using photon count and width of the emission PSF. Once this process is completed, histograms are made for the X and Y dimensions of the selected single-molecule data of the INM protein in question. The rotational symmetry of the NPC usually allows the data to be symmetrically distributed at the NPC's cross-sectional plane, and therefore it is possible to fit the histograms with Gaussian functions to further precisely determine the X and Y centers. Using these centers, the single-molecule INM protein data is adjusted for

the newly determined central axis of the NPC. A visual representation of this process can be found in Supplementary Fig. 5.

**Post-localization 2D to 3D transformation algorithm**. The detailed transformation process used to determine the 3D probability density maps for INM proteins transiting through the NPC can be found in ref. [51,52] and demonstrated again here in Supplementary Fig 3. In short, 3D spatial locations of randomly diffusing molecules inside the NPC can be coordinated in either a Cartesian (x, y, z) or cylindrical coordination system (R, Θ, Y) due to the cylindrical rotational symmetry of the NPC. The 3D molecular locations of INM proteins in the NPC are projected onto a 2D plane in a Cartesian coordination system (x, y) by microscopy imaging. Further projecting onto the x dimension shows that x dimension histograms from either the xy plane and xz plane are identical. Knowing this property, we can create an area matrix in the radial dimension (xz as shown in the figure with concentric rings) such that each column ($A_j$) from the x dimension histogram in the xy plane will be equal to the areas ($s_{i,j}$) times the densities ($\rho_i$) for each radial bin in the xz plane. Finally, the densities in the radial dimension can be determined by solving the matrix equation

$$\left(A_j = 2 * \sum_{i=j}^{n} \rho_i * s_{i,j}\right) \quad (4)$$

which can be used to reconstitute the 3D super-resolution information. The step-to-step demonstration of the data analysis is detailed in the Supplementary Note 1.

**Route localization precision**. To ensure a high reproducibility of 3D spatial probability density maps obtained for each membrane protein candidate, extensive measurements were conducted by combining experimental data and computational simulation. It is important to note that route localization precisions are different from single-molecule localization precision. In detail, the route localization precision is determined by two parameters: one is the number of single-molecule locations and the other is single-molecule localization precision. As shown in Supplementary Fig. 10, simulated data was used to estimate the minimum number of single-molecule localizations required to generate a reliable 3D probability density map for routes of 25 nm (central channel transport) or 40 nm (peripheral channel transport) radial distances. A single-molecule localization precision of 10 nm was used to reflect the precision of our experimentally collected data. We used three different sample sizes (100, 200, and 500 points) and converted the 2D data to 3D probability density map by using our transformation algorithm. Peak positions were fitted for data generated from each of the three sample sizes. 100 fits were used to determine the peak position and the standard deviation is used for the route localization precision. Simulation code can be found at: https://github.com/andrewruba/YangLab.

**Ensemble FRAP by using confocal microscopy**. Ensemble FRAP experiments were performed by using a Leica TCS Sp5 confocal microscope. The microscope is equipped with an HCX PL APO CS oil-immersion objective (×100 and 1.4NA), a 100-W Argon 488-nm laser and the TCS SL software. First, the initial fluorescence intensity value was obtained by averaging the first five pre-bleach images. Then, an Argon laser (488-nm laser line) at full power for about 5 s was used to bleach an area of about 5 μm². Next, fluorescence recovery was measured every 5 s till the fluorescence reached a plateau stage. Finally, by using the FRAP Profiler ImageJ plug-in, image-induced photobleaching was corrected by normalizing to the time-course decay of fluorescence in non-bleached areas. Six biological replicates were used ($n = 6$).

**Single-molecule FRAP measurements**. smFRAP data was first analyzed by using the ImageJ plugin ThunderSTORM (zitmen.github.io/thunderstorm/), then further filtered with a high SNR, and finally their spatial localization positions were determined by GLIMPSE. Moreover, the resultant data was fit with Gaussian functions to determine the distribution of INM proteins on the NE[53]. Finally, the range of INM protein distribution on each of the two membranes was determined by the full width at half maximum (FWHM) of each peak.

**TR calculations**. Using the smFRAP technique, the concentration ratio of LBR and LBR mutants were determined in HeLa cells. From the smFRAP determined ratios, and the total number of LBR on the nuclear envelope, the TR was calculated with the following formulae:

$$R = \frac{N_T * a_2 * A}{a_3 * 2\tau_{1/2}} \quad (5)$$

$$a_1 * F_{mi} = A(a_2 * F_{mo}) + B(a_1 * F_{mi}) \quad (6)$$

$$\frac{A}{B} = \frac{a_2 * F_{mo} * D_o}{a_1 * F_{mi} * D_i} \quad (7)$$

where $N_T$ is the total INM protein molecules found in the cell, $a_1$, $a_2$, and $a_3$ are the INM value, the ONM value and the added INM and ONM value from the INM:

ONM ratio respectively ($a_1 = 3$, $a_2 = 1$, and $a_3 = 3 + 1$ in the case of wild-type LBR). The variable $A$ represents the fraction of INM proteins on the ONM that translocate into the INM after FRAP experiments and $B$ is the fraction of INM proteins on the INM that diffuse into the photobleached area after FRAP. $F_{mi}$ is the mobile fraction on the INM, $F_{mo}$ is the mobile fraction on the ONM. $D_o$ and $D_i$ are the diffusion coefficients on the ONM and INM respectively as determined by single-molecule experiments, and finally, $\tau_{1/2}$ is the time it takes for half fluorescence recovery during FRAP experiments. In the case of LBR we used previously published data for the number of LBR molecules in a cell (~150,000) and the number of NPCs per cell (~2000). Full descriptions of the calculations and the smFRAP technique can be found in ref. [53].

**Bioinformatic analysis**. Forty-six previously confirmed INM localizing proteins and 11 confirmed ONM localizing proteins were used to conduct the bioinformatics analysis. All were confirmed by super resolution OMX microscopy and/or immunogold EM[19,80–83]. First, the primary sequences of these proteins were analyzed for strong NLS sequences in SeqNLS[84], and results with a score higher than 0.86—a high cutoff value suggested by the program developers for more accurate predictions—were classified as sufficiently strong NLSs. Next, the primary sequence was analyzed using MetaDisorder[85] to search for long intrinsically disordered regions of the NET. The structural scaffolding of the NPC is estimated to be ~20 nm and we estimated that it would take roughly 65 amino acids predicted to be intrinsically disordered within a single nucleoplasmic domain for a protein to extend into the central channel. This number was calculated using an average amino acid backbone length of 0.45 nm based on the persistence length of 0.35–0.55 nm for the smallest rigid component of the polypeptide chain[59]. The result from the MetaDisorder analysis was cross-referenced with both Chou-Fasman and Garnier-Robson predictions[58], which look for secondary structure. Only regions with predicted disorder by MetaDisorder that also did not have predictions of structure by the Chou-Fasman and Garnier-Robson algorithms were determined to be intrinsically disordered. Finally, candidate proteins that met both of these requirements and still presented NLS and ID regions that made sense in a biologically relevant manner for facilitated transport through the center of the NPC (e.g., NLS to be recognized by a transport receptor, followed by a sufficiently long ID region to facilitate transit through the physical structure of the NPC, followed by a transmembrane domain to keep the protein tethered to the NE), were categorized as "Both."

**Dual-channel co-tracking experiments using SPEED microscopy**. A 50-mW solid-state 488 nm laser (Obis) and a 50-mW solid-state 561 nm laser (Obis) were used to excite EGFP and RFP tagged LBR respectively in the dual-channel co-tracking experiments. Fluorescence from both fluorophores were collected simultaneously by an objective (×100 and 1.4 NA) and split by the Dual View DV2 system (Photometrics), which includes a dichroic mirror (565dcxr, CHROMA). The two channels were further filtered by a 520 ± 15 nm band pass (CHROMA) and a 593 nm long pass (FF01-593/LP-25; CHROMA) filter. Experiments were performed with just the 488 nm laser to ensure that no FRET occurred during co-tracking experiments. Experiments were performed at 2 ms per frame. Co-tracking error was calculated as <20 nm, based on

$$\sqrt{G_p^2 + R_p^2 + A_p^2} \qquad (8)$$

where $G_p$ (<12 ± 3 nm) is the localization precision for EGFP, $R_p$ (<15 ± 2 nm) is the localization precision for RFP, and $A_p$ (~5 nm) is the error in channel alignment.

**Dual-channel single-molecule FRET experimental setup**. Dual-channel single-molecule FRET experiments were set up and performed using a similar setup as the dual-channel co-tracking experiments, however, only the 488 nm laser (Obis) was used to excite the EGFP labeled N-terminus of LBR. The red channel was observed to detect the fluorescence of RFP on the C-terminus of LBR as it transited through the NPC. We observed no FRET during transit through the NPC.

**Visualization of NETs targeting by fluorescent microscopy**. HeLa cells were seeded on coverslips and transfected with 1.6 μg of plasmid DNA with Lipofectamine2000 reagent (Invitrogen). After 48 h post transfection, cells were fixed in 4% PFA for 10 min at room temperature and washed by PBS. Next, cells were permeabilized in 0.5% Triton in PBS for 5 min at room temperature. After washing, cells were stained with DAPI. Finally, samples were mounted with Vectashield medium (Vector Lab), sealed, and images were taken with a Nikon TE-2000 fluorescence microscope equipped with a 1.45 NA ×100 objective and CoolSnapHQ High Speed Monochrome CCD camera (Photometrics), run by Metamorph image acquisition software. Pictures were analyzed by ImageJ software.

**Western blotting**. HeLa cells were transfected with 1.6 μg of plasmid DNA with Lipofectamine2000 (Invitrogen). Cells were collected after 48 h in 2 x Laemmli sample buffer (120 mM Tris-Cl pH 6.8, 20% glycerol, 4% SDS, 10 mM DTT and 0.02% bromophenol blue). After sonication for 5 s (5 cycles with 1 s on and 1 s off), cells were boiled at 90 °C for 10 min. Next, proteins were resolved by SDS-PAGE,

transferred to Nitrocellulose membrane, blocked by 5% milk in TBST buffer and incubated overnight at 4 °C with rabbit primary antibody anti-GFP (Life Technologies, A11122) and mouse anti-γ-tubulin (SIGMA, T6557) both at a concentration of 1:1000. The membrane was then washed with TBST and incubated with fluorescent secondary antibody from LI-COR (IR680 goat anti-rabbit, IR800 donkey anti-mouse) at a concentration of 1:5000 for 1 h at room temperature. Finally, the membrane was washed with TBST and imaged by the Odyssey CLx system.

**Statistical analysis**. Experiments were reported as mean ±standard error of the mean unless otherwise noted. For comparing three groups, a one-way ANOVA was conducted, followed by a Tukey's HSD post hoc analysis. $P \leq 0.05$ was considered statistically significant. At least 10 biological replicates were used for each condition ($n \geq 10$).

**Reporting summary**. Further information on research design is available in the Nature Research Reporting Summary linked to this article.

## Data availability

All raw data for figures and tables in the manuscript or the supplementary materials is available upon requests.

## Code availability

The Slidebook software package was purchased from Intelligent Imaging Innovations. The GLIMPSE software package is a gift from the Gelles Lab (Brandies University). The ImageJ plugin ThunderSTORM was downloaded from zitmen.github.io/thunderstorm/. Simulation source code has been provided at: https://github.com/andrewruba/YangLab.

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

## Acknowledgements

We would like to thank Professor Glenn Morris (Keele University, UK) for providing human LBR clone and Professor Roland Foisner (Medical University of Vienna, Austria) for wild-type human Lap2B plasmids. We would also like to thank Professor Bettina Buttutaro (Lewis Katz School of Medicine at Temple University) for the use of her confocal microscope. The project was supported by grants from the US National Institutes of Health (NIH GM094041, GM097037, GM116204 and GM22552 to W.Y.) and from the Wellcome Trust (Wellcome Senior Research Fellowship 095209 to E.C.S. and Wellcome 092076 to the Wellcome Centre for Cell Biology).

## Author contributions

K.C.M., E.C.S., and W.Y. designed experiments; K.C.M., Y.L., and W.L. performed microscopy experiments; K.C.M. and R.C. prepared plasmids and established cell lines; K.C.M., S.L.J., Y.L., W.L., C.N., V.O., and W.Y. conducted microscopy data analysis; A.R. wrote the simulation software; K.C.M. and A.R. performed simulations; K.C.M. and E.S. conducted bioinformatic data analysis; K.C.M., R.C. S.L.J., E.S., and W.Y. wrote the manuscript.

## Competing interests

The authors declare no competing interests.
