## [Peer Review File · Nature Communications]

Reviewers' comments:

Reviewer #1 (Remarks to the Author):

Mudumbi et al combine a toolbox of advanced microscopy methods aiming to resolve the long-lasting debate about the nature of different INM protein groups in the peripheral channels in the NPC and their integration within the nuclear pore membrane.

Overall, I find the manuscript exciting, well written and presented (professionally presented Figures), technical sound, and important to be published. I therefore recommend to accept the manuscript for publication in Nature Communications after addressing the following minor comments.

(1) Could the authors please discuss the impact of the mechanical rigidity of INM proteins in the revised version of the manuscript? I appreciate the significance of their length/height. However, the reaction diffusion dynamics and their hydrodynamic radius will also depend on the mechanical status of the molecules.

(2) I think that the manuscript would be strengthened if the authors could show dynamic data, which allows the readership to judge the quality of the measurements. The post-processed data and presentation is very well executed but the raw data acquisition is difficult to judge.

(3) Could the authors please provide statistical significance of the different mobilities as measure by smFRAP in Table 1? Table 1 provides standard deviations but no p-values.

(4) I believe that Nat Com does not allow abbreviations in the abstract.

(5) Could the authors please rephrase the sentence starting with "Here" (line 87)? The reader could be confused that the summary paragraph of the end of the introduction starts at this location, which only follows line 130.

Reviewer #2 (Remarks to the Author):

In this paper Mudumbi et al, show exciting data providing structural evidence how transmembrane proteins are imported via the Nuclear Pore Complex to the inner nuclear membrane. With their advanced and unique single-molecule techniques they make an important contribution to the understanding of the import pathway of several membrane proteins.

The import route that membrane proteins take has been long debated. While the relevance of sorting signals, RanGTP, and NTRs have all been a subject of debate, the fiercest debates concerned the transport route through the NPC. First, multiple labs provided support for the idea that most yeast and human INM proteins use the peripheral channels observed in several EM studies. However, this transport route through the lateral channels was never directly shown. In addition to this canonical route, two yeast INM proteins were proposed to take a different route: they use an active import mechanism which involves the passage of their NTR-bound end of the molecule through the central channel, while their transmembrane domains and the C-terminus travel the peripheral channels. The proposed transport route implied that, at least transiently, openings must exist between the space immediately aligning the pore membrane and the central channel. Only indirect evidence was available so far to support this transport route. According to many in the field, this proposed transport route was incompatible with the current structural data, and hence regarded highly unlikely. Another concern was that it could be an oddity of these two proteins, and/or an oddity of yeast.

In the current manuscript this longstanding debate about where these proteins translocate is addressed using suitable and sophisticated single molecule methods developed by Prof Yang and coworkers and using a range of native proteins. Five human inner nuclear membrane proteins are studied (NET51, NET59, NET99, LBR, Lap2beta). The existence of the peripheral channel as well as passage of Net51/59/99 via the peripheral channels is confirmed. The overall dimensions of the NPC scaffold and the peripheral channels are in good agreement with EM studies, which provides confidence in the accuracy of the SPEED method (which I cannot evaluate in detail). Most exciting is the data on LBR and Lap2beta which show that their NLS and disordered regions pass the central channel, at the same place as soluble proteins with transportins, while the transmembrane and membrane proximal regions pass through the lateral channels. Most convincing are those measurements where the LBR proteins is simultaneously tagged on both termini, so that the recordings of transport through the lateral channels and the central channel reflects one and the same molecule. These measurements provide the first direct evidence that the NPC can accommodate for the hotly debated transport route through the central channel. In fact, the studies by Mudumbi et al align well with the emerging view that the structure of the NPC is much more dynamic than previously thought. In a set of elegant studies the authors additionally show how perturbations of the NPC channels or the sorting signals can alter the transport path of a membrane proteins. This is a surprising finding that may turn out to have important biological implications. Also, the authors provide kinetic information about both transport routes, which is roughly in line with previous estimates from yeast and human systems. In addition to the experimental data the authors attempt to address the question how prevalent both transport mechanisms are. They estimate that only some 10% have ID linkers and NLSs.

This paper is in my opinion an important and novel paper that would be very suitable for publication in nature communications after a few controls and textual changes are considered.

Main concerns:

1) the dual-channel co-tracking SPEED microscopy of the double tagged LBR is important since it shows that single molecules of LBR pass the NPC as a full-length protein and membrane embedded. Similar experiments are not included for Lap2 β . This is a potential problem in the interpretation of the data for Lap2 β , as, obviously, the presence of truncated versions encoding only the nucleoplasmic, soluble domain of LBR could easily explain the signal the central channel. I realize that doing dual channel co-tracking of Lap2 β would involve a very significant extra effort and this is not required in my opinion. I would however like to see western blot and microscopy images of the cells expressing the lagged Lap2 β , so that readers can be convinced that the data very likely reflects full length proteins and not a degradation product of Lap2 β .

2) The estimate that ~9% of the inner nuclear membrane proteins may encode an ID domain and an NLS is really only an estimate as should be referred to as such (also in the abstract). The assumption that all these "extend intrinsically disordered (ID) domains containing nuclear localization signals (NLSs) into the central channel for directed nuclear transport" is also really only an assumption and should be referred to as such. To clarify the analysis done, I have three more suggestions: Please provide the list of 46 INM and 11 ONM proteins that were included in the analysis. Line 190 (methods) Please indicate the rationale for taking SeqNLS scores higher than 0.86 as 'sufficiently strong' NLSs. Please elaborate what "made sense in a biological manner for facilitated transport through the center of the NPC " means.

Minor comments

3) I have few suggestions towards placing the work in the context of the literature.

a. The two papers that were published in 2015, by the Ellenberg and by the Kutay lab (Boni et al, JCB 2015, Ungricht et al JCB 2015) should be discussed. Both groups used life imaging and computational

techniques to show that a diffusion-retention mechanism can explain the import of INM proteins. In both papers Lap2 β and LBR are studied, like in this manuscript. The estimates of transport kinetics in Boni et al match nicely the ones reported here (5.6 molecules per minute per NPC versus 4.6 molecules/NPC/min), but while the numbers match, the conclusion of Boni et al is that their data supports a diffusion retention model for INM protein transport.

b. In the introduction Line 107-109 is an incorrect statement: In Meinema et al (2011) the endogenous ID region of the yeast proteins was used and in Kralt et al. MolBiol Cell 2015 the full length and endogenously expressed protein (native promotor, native chromosomal location) was used to study the NLS. If the authors meant to say that that thus far there were no papers that studied the relevance of an ID linker in the context of the full length and natively expressed protein, that would be correct.

c. In the Discussion: Line 341-343. The experiments do not address this concern directly (different proteins, different cells). Laba et al (Cells, 2015) confirms the transmembrane nature of the Heh2-derived truncated proteins.

3) Line 255: there is no data to support that the R74T mutation indeed results in the disruption of NTR binding. Is there biochemical evidence to support this statement? Or is there data to show mislocalisation of GFP-NLS(R74T) fusions in cells? Ma et al 2007 J Cell Sci can be cited to support that imp beta is bound.

4) Other:

Line 220-223: Lap2 β and Lap β are used both throughout the manuscript.

Line 261: Lap β Δ NLS: actually the NLS is not deleted but mutated: R319T and R320T. Please mention.

Line 160 (methods) range not rage.

Line 181 (methods) In the case of LBR it reads that the no of copies per cell was based on a previous estimate; what was used for the other proteins?

Line 218-223 (methods). I do not understand which figures or experiments this related to.

Line 265-268: this paragraph is slightly confusing. Maybe add: ...and roughly 80% of the length of the nucleoplasmic region still present, LBR required these ID Linker regions to transit....etc.

Smovie 2 does not work

Reviewer #3 (Remarks to the Author):

This paper has studied how transmembrane proteins in their inner nuclear membrane are transported to the inner nuclear membrane using live cell high-speed super-resolution single-molecule microscopy. Authors found that either just the peripheral channels or both central and peripheral channels together are used for the transport depending on their nucleoplasmic signals. They succeeded in visualizing pathways and single-molecule trajectories going through the central and peripheral channels distinguishing the positions of the central and peripheral channels and pore membrane lumen. However, the paper has substantive issues in description, which makes the paper hard to comprehend.

Major:

1. There is a problem with the logical description. In addition, some paragraphs are too long in the Results and Introduction sections. The Discussion section has insufficient contents.

2. Abstract is inappropriate as summary. It does not reflect the entire contents, and combining Abstract and Significance Statement makes a better summary. Furthermore, for example, the value of “~9%” is described only once in the Results section, and Fig. 4E is referred. However, Figure 4E and its legend has no description of the clear basis for calculation, and the accuracy of the numerical value of 9 is not described.

3. The title is not suitable. The word “Nucleoplasmic signals” of the title subject is used only once in

the Significance Statement section through the whole manuscript, and no explanation is described even though the word is not in common use. The word “simultaneously” is also used, but it is not clear with which nucleoplasmic signals promote simultaneously.

4. It is written that the route precision is ~2 nm at line 164 in the Result section, and 0.90 ~ 1.42 in Table S2 based on the simulations (Figs. S8-10). However, the validity of using the values is not clear. It is considered appropriate to use the standard deviation of the distribution or that of data from independent measurements (not standard error of the mean SEM; more precisely, Student's t value multiplied by SEM) as the precision and error of the route and diameter.

Minor:

1. The sentence “~9% additionally extend ...” in Abstract is unclear, since the subject of the sentence is not clear.

Reviewer #1:

Mudumbi et al combine a toolbox of advanced microscopy methods aiming to resolve the long-lasting debate about the nature of different INM protein groups in the peripheral channels in the NPC and their integration within the nuclear pore membrane.

Overall, I find the manuscript exciting, well written and presented (professionally presented Figures), technical sound, and important to be published. I therefore recommend to accept the manuscript for publication in Nature Communications after addressing the following minor comments.

>> We thank the Reviewer for their strong support of our work.

(1) Could the authors please discuss the impact of the mechanical rigidity of INM proteins in the revised version of the manuscript? I appreciate the significance of their length/height. However, the reaction diffusion dynamics and their hydrodynamic radius will also depend on the mechanical status of the molecules.

>> We have rewritten this part of the introduction to explain this more clearly and note the issue of rigidity and hydrodynamic radius raised. It now reads "*... INM proteins are restricted to the multiple peripheral NPC channels, which cryo-electron microscopy (cryo-EM) suggests are ~10 nm wide (Alber et al., 2007; Hinshaw et al., 1992; Kim et al., 2018; Maimon et al., 2012; Mosalaganti et al., 2018). This would limit the nucleoplasmic domains of INM proteins to ~60 kDa if they had a globular structure since this would yield a hydrodynamic radius of ~10 nm. If their structure were more linear the proteins could still in theory snake through these channels with a smaller radius in the orientation of transport. This 60 kDa limit has been experimentally confirmed (Ohba et al., 2004; Soullam and Worman, 1995) and is characteristic for the wide range of NETs identified in the nuclear envelope (Schirmer et al., 2003; Zuleger et al., 2011). However, most NETs are not likely to have a rigid highly folded structure as most have regions of intrinsic order (ID) within their nucleoplasmic domains.*".

(2) I think that the manuscript would be strengthened if the authors could show dynamic data, which allows the readership to judge the quality of the measurements. The post-processed data and presentation is very well executed but the raw data acquisition is difficult to judge.

>> As per the Reviewer's suggestion and the journal's policy, all raw data available upon request. Furthermore, we also provided the raw data for Movie 2 (a raw movie in black/white without a post process) named as Movie 2A in the resubmitted manuscript.

(3) Could the authors please provide statistical significance of the different mobilities as measure by smFRAP in Table 1? Table 1 provides standard deviations but no p-values.

>> We would like to thank the Reviewer for pointing out this oversight, the appropriate statistics along with F and p-values have been provided in Table 1.

(4) I believe that Nat Com does not allow abbreviations in the abstract.

>> Thank you for pointing this out as we had not seen it in the guidelines. We have now removed the abbreviations from the abstract.

(5) Could the authors please rephrase the sentence starting with "Here" (line 87)? The reader could be confused that the summary paragraph of the end of the introduction starts at this location, which only follows line 130.

>> As suggested, to avoid confusion, we have rephrased the sentence at line 87 to read: This study assesses two NPC-dependent models: free lateral diffusion-retention and nuclear localization signal (NLS)-dependent facilitated transport.

Reviewer #2:

In this paper Mudumbi et al, show exciting data providing structural evidence how transmembrane proteins are imported via the Nuclear Pore Complex to the inner nuclear membrane. With their advanced and unique single-molecule techniques they make an important contribution to the understanding of the import pathway of several membrane proteins.

The import route that membrane proteins take has been long debated. While the relevance of sorting signals, RanGTP, and NTRs have all been a subject of debate, the fiercest debates concerned the transport route through the NPC. First, multiple labs provided support for the idea that most yeast and human INM proteins use the peripheral channels observed in several EM studies. However, this transport route through the lateral channels was never directly shown. In addition to this canonical route, two yeast INM proteins were proposed to take a different route: they use an active import mechanism which involves the passage of their NTR-bound end of the molecule through the central channel, while their transmembrane domains and the C-terminus travel the peripheral channels. The proposed transport route implied that, at least transiently, openings must exist between the space immediately aligning the pore membrane and the central channel. Only indirect evidence was available so far to support this transport route. According to many in the field, this proposed transport route was incompatible with the current structural data, and hence regarded highly unlikely. Another concern was that it could be an oddity of these two proteins, and/or an oddity of yeast.

In the current manuscript this longstanding debate about where these proteins translocate is addressed using suitable and sophisticated single molecule methods developed by Prof Yang and coworkers and using a range of native proteins. Five human inner nuclear membrane proteins are studied (NET51, NET59, NET99, LBR, Lap2beta). The existence of the peripheral channel as well as passage of Net51/59/99 via the peripheral channels is confirmed. The overall dimensions of the NPC scaffold and the peripheral channels are in good agreement with EM studies, which provides confidence in the accuracy of the SPEED method (which I cannot evaluate in detail). Most exciting is the data on LBR and Lap2beta which show that their NLS and disordered regions pass the central channel, at the same place as soluble proteins with transportins, while the transmembrane and membrane proximal regions pass through the lateral channels. Most convincing are those measurements where the LBR proteins is simultaneously tagged on both termini, so that the recordings of transport through the lateral channels and the central channel reflects one and the same molecule. These measurements provide the first direct evidence that the NPC can accommodate for the hotly debated transport route through the central channel. In fact, the studies by Mudumbi et al align well with the emerging view that the structure of the NPC is much more dynamic than previously thought. In a set of elegant studies the authors additionally show how perturbations of the NPC channels or the sorting signals can alter the transport path of a membrane proteins. This is a surprising finding that may turn out to have important biological implications. Also, the authors provide kinetic information about both transport routes, which is roughly in line with previous estimates from yeast and human systems. In addition to the experimental data the authors attempt to address the question how prevalent both transport mechanisms are. They estimate that only some 10% have ID linkers and NLSs.

This paper is in my opinion an important and novel paper that would be very suitable for publication in nature communications after a few controls and textual changes are considered.

>> We thank the Reviewer for their strong support of our work as well as the suggestions provided which help improve the work and place it in better context with existing literature.

Main concerns:

1) the dual-channel co-tracking SPEED microscopy of the double tagged LBR is important since it shows that single molecules of LBR pass the NPC as a full-length protein and membrane embedded. Similar experiments are not included for Lap2 β . This is a potential problem in the interpretation of the data for Lap2 β , as, obviously, the presence of truncated versions encoding only the nucleoplasmic, soluble domain of LBR could easily explain the signal the central channel. I realize that doing dual channel co-tracking of Lap2 β would involve a very significant extra effort and this is not required in my opinion. I would however like to see western blot and microscopy images of the cells expressing the lagged Lap2 β , so that readers can be convinced that the data very likely reflects full length proteins and not a degradation product of Lap2 β .

>> This is an excellent suggestion and we agree that it is a much needed experiment. As such, we have performed Western blot experiments to show the expression levels of LBR and Lap2 β . Furthermore, as suggested we have also included fluorescence images of all the various LBR and Lap2 β constructs in Supplemental Figure 12.

2) The estimate that ~9% of the inner nuclear membrane proteins may encode an ID domain and an NLS is really only an estimate as should be referred to as such (also in the abstract). The assumption that all these “extend intrinsically disordered (ID) domains containing nuclear localization signals (NLSs) into the central channel for directed nuclear transport” is also really only an assumption and should be referred to as such.

>> We appreciate the suggestion and agree that this is indeed just a prediction based on our bioinformatic analysis and have rephrased the appropriate portions of the manuscript to reflect this. In particular we have taken the ~9% out of the abstract where there was insufficient space to explain the calculation and added the qualifier “apparently” for the latter point. In the text we have made similar minor changes to better qualify these issues.

To clarify the analysis done, I have three more suggestions: Please provide the list of 46 INM and 11 ONM proteins that were included in the analysis. Line 190 (methods) Please indicate the rationale for taking SeqNLS scores higher than 0.86 as ‘sufficiently strong’ NLSs. Please elaborate what “made sense in a biological manner for facilitated transport through the center of the NPC “ means.

>> As suggested, we agree that including a table of the 46 INM and 11 ONM proteins will improve the quality of this paper and have included it as a new table (Table S4).

In regards to the SeqNLS score used, the authors of the software state in their paper (Lin and Hu, 2013):

It shows that when the enrichment-score cutoff is set higher, the precision of the predictor increases. This is because the matches of the sequential patterns with the higher enrichment score are more significant and thus are more likely to be part of NLS.

Therefore, we used the second highest cutoff available in the software, a score of 0.86 (the highest score being 0.89). We have further clarified this in the text.

We made the following changes to the methods section to clarify the confusion pointed out by the Reviewer. The new text reads as follows:

Finally, candidate proteins that met both of these requirements and still presented NLS and ID regions that made sense in a biologically relevant manner for facilitated transport through the center of the NPC (e.g. NLS to be recognized by a transport receptor, followed by a sufficiently long ID region to facilitate transit through the physical structure of the NPC, followed by a transmembrane domain to keep the protein tethered to the NE), were categorized as “Both.”

Minor comments

3) I have few suggestions towards placing the work in the context of the literature.

a. The two papers that were published in 2015, by the Ellenberg and by the Kutay lab (Boni et al, JCB 2015, Ungricht et al JCB 2015) should be discussed. Both groups used life imaging and computational techniques to show that a diffusion-retention mechanism can explain the import of INM proteins. In both papers Lap2 β and LBR are studied, like in this manuscript. The estimates of transport kinetics in Boni et al match nicely the ones reported here (5.6 molecules per minute per NPC versus 4.6 molecules/NPC/min), but while the numbers match, the conclusion of Boni et al is that their data supports a diffusion retention model for INM protein transport.

>> We wholeheartedly agree that our work should be placed in context with these particular studies and have included the following paragraph:

Several recent studies also used in vivo imaging techniques, which took advantage of cleavable sequences to determine INM protein transport (Boni et al., 2015; Ungricht et al., 2015). Using these methods, both groups concluded that LBR and Lap2 β transit into the INM using some form of lateral-diffusion retention. Ungricht and colleagues also conclude that this transit is energy dependent, which is suggestive of transport receptor mediated trafficking. In contrast, based on the INM localization of LBR after knocking down numerous transport receptors, Boni et al suggest that the lateral-diffusion retention model is sufficient to explain INM protein import since the presence of transport receptors do not seem to affect LBR localization. However, as shown by our approach, the loss of NLSs – equivalent to the loss of access to transport receptors – reduces the transport rate of LBR and changes its transport mechanism from the central channel of the NPC to the peripheral channel, but allows it to still localize to the INM. By directly observing INM protein transport and distinguish between the lateral-diffusion retention model and transport receptor mediated transport, we can validate both approaches while providing more nuanced context.

b. In the introduction Line 107-109 is an incorrect statement: In Meinema et al (2011) the endogenous ID region of the yeast proteins was used and in Kralt et al. MolBiol Cell 2015 the full length and endogenously expressed protein (native promotor, native chromosomal location) was used to study the NLS. If the authors meant to say that thus far there were no papers that studied the relevance of an ID linker in the context of the full length and natively expressed protein, that would be correct.

>> We appreciate the correction and have modified the statement to read:

However, previous studies used truncated proteins, and the relevance of endogenous ID regions in full-length genome-encoded NETs remains to be tested.

c. In the Discussion: Line 341-343. The experiments do not address this concern directly (different proteins, different cells). Laba et al (Cells, 2015) confirms the transmembrane nature of the Heh2-derived truncated proteins.

>> We thank the reviewer for pointing this out. We have now changed this sentence to read "*The finding that different regions of the same protein transit simultaneously through both central and peripheral channels enhances previous studies arguing for central channel transport (Meinema et al., 2011; Laba et al., 2015)*". We think the work from the Veenhoff lab was the defining study for this mechanism and had not seen that paper that clarified this one limitation of the original study. We believe that this work still makes an extremely important contribution in showing the simultaneous transport through both channels. The data shown in Figure 3B and C (and trajectories from single-GFP tagged INM proteins – data not shown) directly show that the proteins are tethered in the NE and also tracked from the same protein, as can be seen from the ~18 nm distance between GFP and RFP during transit. Furthermore, both Western Blots and immunofluorescence confirm the transmembrane nature of each INM protein (newly added Fig. S12).

3) Line 255: there is no data to support that the R74T mutation indeed results in the disruption of NTR binding. Is there biochemical evidence to support this statement? Or is there data to show mislocalisation of GFP-NLS(R74T) fusions in cells? Ma et al 2007 J Cell Sci can be cited to support that imp beta is bound.

>> The Reviewer is correct in pointing out that we do not have biochemical evidence for this. Therefore we have changed the sentence to "*The second construct was a point mutation in just one of the four LBR NLS sequences, where an arginine at position 74 of the strongest predicted NLS – also supported by previous studies (Ma et al., 2007) – was mutated to a threonine (LBR R74T), which should in theory create a non-functional NLS while keeping the ID domain intact and the total length of the N-terminus unchanged.*" Nonetheless, the data generated with this construct supports the biochemical function due to the change in transport route and rate as assessed by our SPEED and smFRAP assays and our immunofluorescence assay shows LBR in both the ER and the NE as with other overexpressed proteins.

4) Other:

Line 220-223: Lap2 β and Lap β are used both throughout the manuscript.

>> Thank you for pointing this out. This oversight has been corrected.

Line 261: Lap β Δ NLS: actually the NLS is not deleted but mutated: R319T and R320T. Please mention.

>> The Reviewer is correct to note that the NLS was not deleted but mutated, we believe that the following statement in the manuscript also reflects this:

Similarly, the NLS in the N-terminus of Lap2 β was mutated (two arginines were mutated to theronines, Lap2 β Δ NLS)

Line 160 (methods) range not rage.

>> Thank you for pointing this out. We have corrected this typo.

Line 181 (methods) In the case of LBR it reads that the no of copies per cell was based on a previous estimate; what was used for the other proteins?

>> The copy number of LBR per cell was only used to calculate the translocation rate, and is therefore not used for other proteins where the translocation rate was not determined by smFRAP.

Line 218-223 (methods). I do not understand which figures or experiments this related to.

>> We thank the Reviewer for bringing this to our attention, and we have added the following sentences in the main text for clarity:

Furthermore, since GFP and RFP are FRET pairs, we performed experiments in which the GFP was excited and the RFP channel was observed for FRET. These experiments did not produce a FRET signal, indicating that the GFP and RFP did not both travel through the peripheral channel, where they would be closely confined and give a FRET signal.

Line 265-268: this paragraph is slightly confusing. Maybe add: ...and roughly 80% of the length of the nucleoplasmic region still present, LBR required these ID Linker regions to transit....etc.

>> We appreciate the suggested change for clarity and have added it to the manuscript.

Smovie 2 does not work

>> We have replaced the movie in the resubmitted version.

Reviewer #3

This paper has studied how transmembrane proteins in their inner nuclear membrane are transported to the inner nuclear membrane using live cell high-speed super-resolution single-molecule microscopy. Authors found that either just the peripheral channels or both central and peripheral channels together are used for the transport depending on their nucleoplasmic signals. They succeeded in visualizing pathways and single-molecule trajectories going through the central and peripheral channels distinguishing the positions of the central and peripheral channels and pore membrane lumen. However, the paper has substantive issues in description, which makes the paper hard to comprehend.

>> We have carefully read over the Reviewer suggestions as well as the manuscript and have reworded and revised the manuscript for clarity and comprehension. We believe that this new version addresses the concerns of the Reviewer and will allow the work to be accessible for a large audience of varied expertise.

Major:

1. There is a problem with the logical description. In addition, some paragraphs are too long in the Results and Introduction sections. The Discussion section has insufficient contents.

>> We thank the Reviewer for their feedback and have made several changes to the results and introduction sections for clarity and added extensive context to the discussion to place our work in context of previous research.

2. Abstract is inappropriate as summary. It does not reflect the entire contents, and combining Abstract and Significance Statement makes a better summary. Furthermore, for example, the value of “~9%” is described only once in the Results section, and Fig. 4E is referred. However, Figure 4E and its legend has no description of the clear basis for calculation, and the accuracy of the numerical value of 9 is not described.

>> As suggested by the Reviewer, we have re-written our abstract and incorporated many of the ideas from the significance statement (which has now been removed) into the new version to improve clarity.

3. The title is not suitable. The word “Nucleoplasmic signals” of the title subject is used only once in the Significance Statement section through the whole manuscript, and no explanation is described even though the word is not in common use. The word “simultaneously” is also used, but it is not clear with which nucleoplasmic signals promote simultaneously.

>> We appreciate the Reviewers' concern, but are limited by word constraints in the title. Therefore, we have instead defined both in the abstract and in the introduction what we mean by nucleoplasmic signals as follows:

Abstract - "...finding that most inner nuclear membrane proteins use only the peripheral channels, but some apparently extend intrinsically disordered domains containing nuclear localization signals into the central channel for directed nuclear transport. These nucleoplasmic signals are critical for..."

Introduction - "Several nucleoplasmic signals – signals such as nuclear localization signals and these ID regions found on the nucleoplasmic domain of INM proteins – have been shown to be important for transport. For instance, transport receptors that have been shown to facilitate central channel transport (Bayliss et al., 2000, 2002; Otsuka et al., 2008; Radu et al., 1995) are too big to fit into the peripheral NPC channels, yet transport receptors importin alpha and beta were shown to be important for NET transport in yeast (King et al., 2006; Meinema et al., 2011). Thus, the NLS-dependent mechanism posits long ID regions in the nucleoplasmic domains of INM proteins that can reach through the NPC core structure into the central channel (~50 nm wide at the narrowest region) to enable the NLSs to bind transport receptors and phenylalanine-glycine (FG) Nups, similar to transport of soluble proteins (King et al., 2006; Meinema et al., 2011)"

4. It is written that the route precision is ~2 nm at line 164 in the Result section, and 0.90 ~ 1.42 in Table S2 based on the simulations (Figs. S8-10). However, the validity of using the values is not clear. It is considered appropriate to use the standard deviation of the distribution or that of data from independent measurements (not standard error of the mean SEM; more precisely, Student's t value multiplied by SEM) as the precision and error of the route and diameter.

>> As the Reviewer astutely points out, this is a typo that should read that the route precision is < 2 nm. This has been fixed in the text. The Reviewer correctly states that the standard deviation of the distribution should be used as the route precision. We do just that, as stated in both Table S2, Figs. S8-10, and the methods. Succinctly, the route precision (σ_{TR}), refers to the standard deviation of the mean fitting of histograms composed of mean radius fittings from ≥ 100 simulations.

Minor:

1. The sentence “~9% additionally extend ...” in Abstract is unclear, since the subject of the sentence is not clear.

>> We thank the Reviewer for bringing this to our attention and we have removed this from the abstract as there was not sufficient space to further explain it and thus removed the problem.

REVIEWERS' COMMENTS:

Reviewer #1 (Remarks to the Author):

The authors addressed satisfactorily my previously raised concerns, and therefore recommend to accept the manuscript for publication in Nature Communications.

Reviewer #2 (Remarks to the Author):

My concerns have been dealt with satisfactory and I support publication. I have one remaining suggestions:

To show the complete western in S13 and not just a section reflecting where the full length protein runs, as this control is meant to take away the concern of (soluble) breakdown products complicating the interpretation of the data.

REVIEWERS' COMMENTS:

Reviewer #1 (Remarks to the Author):

The authors addressed satisfactorily my previously raised concerns, and therefore recommend to accept the manuscript for publication in Nature Communications.

>> Thanks for the reviewer's recommendation and his/her effort in reviewing our manuscript.

Reviewer #2 (Remarks to the Author):

My concerns have been dealt with satisfactory and I support publication. I have one remaining suggestions:

To show the complete western in S13 and not just a section reflecting where the full length protein runs, as this control is meant to take away the concern of (soluble) breakdown products complicating the interpretation of the data.

>> Thanks for this reviewer's effort and recommendation. As requested, we have added the complete western gel into the Supplementary Figure.